# The *Anopheles gambiae* 2La chromosome inversion is associated with susceptibility to *Plasmodium falciparum* in Africa

**Michelle M Riehle[1], Tullu Bukhari[2], Awa Gneme[3], Wamdaogo M Guelbeogo[3], Boubacar Coulibaly[4], Abdrahamane Fofana[4], Adrien Pain[5,6,7], Emmanuel Bischoff[5,6], Francois Renaud[8,9], Abdoul H Beavogui[10], Sekou F Traore[4], N'Fale Sagnon[3], Kenneth D Vernick[5,6]***

[1]Department of Microbiology and Immunology, University of Minnesota, Minneapolis, United States; [2]Department of Zoology, Maseno University, Maseno, Kenya; [3]Centre National de Recherche et de Formation sur le Paludisme, Ouagadougou, Burkina Faso; [4]Malaria Research and Training Centre, Faculty of Medicine and Dentistry, University of Mali, Bamako, Mali; [5]Department of Parasites and Insect Vectors, Unit of Genetics and Genomics of Insect Vectors, Institut Pasteur, Paris, France; [6]CNRS Unit of Hosts, Vectors and Pathogens, Paris, France; [7]Bioinformatics and Biostatistics Hub (C3BI), USR 3756 IP CNRS, Institut Pasteur, Paris, France; [8]Laboratoire Maladies Infectieuses et Vecteurs: Ecologie Génétique, Evolution et Contrôle (MIVEGEC), Institut de Recherche pour le Développement (IRD), Montpellier, France; [9]UMR 224-5290, Centre National de la Recherche Scientifique (CNRS), Montpellier, France; [10]Centre de Formation et de Recherche en Santé Rurale de Mafèrinyah, Conakry, Guinea

**\*For correspondence:** kvernick@ pasteur.fr

**Competing interests:** The authors declare that no competing interests exist.

**Abstract** Chromosome inversions suppress genetic recombination and establish co-adapted gene complexes, or supergenes. The 2La inversion is a widespread polymorphism in the *Anopheles gambiae* species complex, the major African mosquito vectors of human malaria. Here we show that alleles of the 2La inversion are associated with natural malaria infection levels in wild-captured vectors from West and East Africa. Mosquitoes carrying the more-susceptible allele (2L+$^a$) are also behaviorally less likely to be found inside houses. Vector control tools that target indoor-resting mosquitoes, such as bednets and insecticides, are currently the cornerstone of malaria control in Africa. Populations with high levels of the 2L+$^a$ allele may form reservoirs of persistent outdoor malaria transmission requiring novel measures for surveillance and control. The 2La inversion is a major and previously unappreciated component of the natural malaria transmission system in Africa, influencing both malaria susceptibility and vector behavior.

## Introduction

Mosquitoes of the *Anopheles gambiae* species complex are primary African vectors of the human malaria parasite, *Plasmodium falciparum*, which is responsible for extensive human morbidity and mortality. At least part of the widespread geographic success of *A. gambiae* and its sister taxon *A. coluzzii* (previously called the *A. gambiae* S and M molecular forms, respectively [*Coetzee et al., 2013*]) can be attributed to segregating paracentric chromosomal inversions. Inversions are physical rearrangements of a chromosome segment that suppress recombination and limit exchange of variation between two alternate suites of alleles. These diverged adaptive gene complexes function as

supergenes in many organisms (*Thompson and Jiggins, 2014*) and influence diverse phenotypes including butterfly color mimicry (*Joron et al., 2011*; *Nishikawa et al., 2015*), ant colonial behavior (*Wang et al., 2013*), avian reproductive morphs (*Lamichhaney et al., 2016*; *Huynh et al., 2011*), human fertility (*Stefansson et al., 2005*), and *Drosophila* ecological adaptation (*Corbett-Detig and Hartl, 2012*; *Balanyá et al., 2006*). In African *Anopheles*, inversion polymorphism is thought to enable fine-grained adaptation of malaria vectors to diverse ecological conditions, thus expanding the range of an important malaria vector throughout Africa (*Coluzzi et al., 1979*).

The 2La inversion comprises ~10% of the genome size and segregates in *A. gambiae* and *A. coluzzii*, the only species of the complex in which it is polymorphic. Field-based studies have found no inherent difference in *P. falciparum* susceptibility between these main vector species of the *A. gambiae* complex, *A. gambiae* and *A. coluzzii* (*Fryxell et al., 2012*; *Gnémé et al., 2013*). Polymorphism of the 2La inversion has been correlated with aspects of vector bionomics, particularly adaptation to aridity or humidity (*Coluzzi et al., 1979*; *Fouet et al., 2012*), and biting and resting behavior (*Coluzzi et al., 1977*). The two forms of the inversion are allelic, by convention designated as 2L+[a] and 2La. The inversion polymorphism is widespread in Africa, but the alleles are not evenly distributed across geography. An allelic frequency cline was described in West Africa, where high frequencies of 2L+[a] are present in humid southern forest and coastal regions, with the 2La allele gradually becoming predominant northward, until becoming essentially fixed in the arid sub-Saharan savanna (*Coluzzi et al., 1979*; *Touré et al., 1998*). Subsequent studies have tended to confirm a correlation of the 2L+[a] allele with ecological humidity (*Cheng et al., 2012*).

A prior cytogenetic study detected differences in *P. falciparum* infection rates of chromosome inversion karyotypes of *A. gambiae* in Kenya (*Petrarca and Beier, 1992*). In that report, mosquitoes homozygous for the 2L+[a]/2L+[a] karyotype displayed a two-fold higher prevalence of *P. falciparum* infection than sympatric inverted 2La/2La homokaryotypes. However, as observed in that work, the intraspecific differences in infection rate required confirmation at other geographic sites to generalize the findings and could potentially be due to many interacting factors, including differential longevity, physiological susceptibility to *Plasmodium*, and feeding behavior. To our knowledge, association of the 2La inversion with *Plasmodium* infection has only been examined once since the initial cytogenetic study, and no significant effect was detected (*Matoke-Muhia et al., 2016*), but the small sample size of infected mosquitoes in the study (n = 15) would yield low statistical power to detect an effect among three genotypes, even if one existed.

Vector genetic heterogeneity for parasite transmission is an important question during the malaria pre-elimination stage (*Brady et al., 2016*). Reservoirs of highly efficient vectors, or vectors that evade control, can contribute unevenly and cryptically to residual malaria transmission. After implementing existing vector control approaches such as insecticide-treated bednets (ITNs), indoor residual insecticide spraying (IRS), and drug treatment of symptomatic human cases, it may be necessary to develop specialized tools to detect and control persistent sources of residual transmission.

In the current study, we detect differential *P. falciparum* infection rates among carriers of 2La inversion genotypes at study sites across the African continent, and we test potential hypotheses regarding the underlying mechanism of the infection phenotype. A single evolutionary origin of the 2La inversion across geography and *A. gambiae* taxa indicates that the differential infection phenotype probably results from a shared, common mechanism rather than multiple local ones. We dissect the influence of multiple epidemiological factors, including vector blood-feeding behavior, survival rates, and general antimicrobial immune competence, to provide information about potential factors underlying the phenotype. Taken together, the results point toward a mechanism that includes physiological differences for *P. falciparum* susceptibility encoded by the 2La inversion haplotypes. We discuss possible candidate genes in the genomic region, as well as the limitations of genetically resolving a trait encoded by long non-recombining inversion haplotypes. We also examine correlation of 2La inversion genotype with mosquito feeding and resting behaviors, which has implications both for infection epidemiology and vector control strategies. Finally, we correlate the nonrandom geographic distribution of 2La inversion alleles with ecological conditions important for malaria transmission. This work describes and characterizes a novel heterogeneity of the natural malaria transmission cycle present across the African continent, with potentially significant effect on transmission dynamics as well as vector control strategies.

## Results

### The 2La inversion is significantly associated with differential malaria infection

Infection rates of *P. falciparum* were measured in mosquitoes of the *A. gambiae* species complex in the Western Highlands of Kenya, in the forested zone of southern Republic of Guinea (Guinea-Conakry), and in the Sudan-Savanna zone of Burkina Faso (*Figure 1*). The infection samples were generated by three distinct infection regimes, in order to control for potential biases of any individual method: (i) wild-caught mosquitoes that took a bloodmeal, if at all, under entirely natural conditions and were dissected after holding for 1 week in the insectary (Kenya) to measure infections acquired in nature and developed under controlled conditions, or (ii) wild-caught and dissected within 2 days of capture (Guinea-Conakry) to measure infections acquired and developed entirely in nature; or (iii) mosquitoes challenged with wild gametocytes by membrane feeding under controlled conditions (Burkina Faso). Exposure of the wild-caught mosquitoes (Kenya and Guinea-Conakry) to parasites occurred freely in nature, and thus infection outcome is the combined effect of intrinsic genetic susceptibility plus the influence of factors such as human or animal blood-feeding preference, mosquito age, previous infection history, influence of resting-site preference, and other uncontrolled environmental variables. In contrast, the experimentally infected samples control for all behavioral and environmental factors that could influence mosquito infection in nature, and measure intrinsic genetic susceptibility.

Numbers of *P. falciparum* midgut oocysts were determined by dissection and microscopy, and genotypes of the 2La inversion were detected in the same individuals by molecular genotyping. We analyzed the association of 2La inversion genotype with *P. falciparum* infection outcome at sites where all three genotypes of the inversion segregate in sympatry. All comparisons were made between 2La genotypes within geographic sites such that only sympatric genotypes are compared and were not made across sites.

In an analysis of oocyst infection prevalence, *A. gambiae* complex mosquitoes with the homozygous $2L+^a/2L+^a$ inversion genotype were significantly more likely to carry midgut oocysts than the 2La/2La homozygotes at all three study sites (*Figure 2A*, Chi-squared = 11.25, df = 2, p=0.004 for Kenya, Chi-squared = 7.96, df = 2, p=0.02 for Guinea-Conakry and Chi-squared = 33.52, df = 2, p=$5*10^{-8}$ for Burkina Faso). Post-hoc pairwise tests show that the 2La/2La homozygotes were significantly less likely to be infected than either heterozygotes or $2L+^a/2L+^a$ homozygotes (*Figure 2—figure supplement 1*).

For the analysis of infection prevalence in *Figure 2A*, test sets included all infected mosquitoes from each collection, and an equal number of randomly chosen uninfected mosquitoes from the same collection (see Materials and methods for greater detail). The test sets thus display an average infection prevalence of 50%, and have statistical power to detect genetic association for either increased or decreased infection prevalence. This analytical approach normalizes statistical power to query genetic effects in both phenotypic directions across heterogeneous sample sets, which would otherwise display different levels of statistical power if not normalized.

Association of oocyst infection intensity and 2La genotype was also tested (*Figure 2B*). Infection intensity is defined as the number of midgut oocysts in mosquitoes with ≥1 oocyst, because including uninfected mosquitoes would confound infection prevalence with intensity. In the Burkina Faso samples, infection intensity was significantly higher in the $2L+^a/2L+^a$ homozygotes as compared to 2La/2La homozygotes (p=0.01). The samples from Kenya displayed a trend in the same direction, although non-significant. The Burkina Faso sample set was larger than the Kenya samples, which may generate greater statistical power. The samples from Guinea-Conakry did not have statistical power for analysis of infection intensity because all seven 2La/2La homozygotes were uninfected (*Figure 2—figure supplement 1*).

Finally, infection association was analyzed as the proportion of each 2La inversion haplotype present in the infected and uninfected groups at each geographic study site (*Figure 2C*). If the 2La inversion genotype has no influence upon infection outcome, then inversion frequencies should be randomized among uninfected and infected mosquitoes. However, 2La/2La homozygotes were present at higher frequency in uninfected mosquitoes, while $2L+^a/2L+^a$ homozygotes were at higher frequency in infected mosquitoes. All the results are significant, in the same direction, and with similar p-values, as in the infection prevalence analysis (*Figure 2A*).

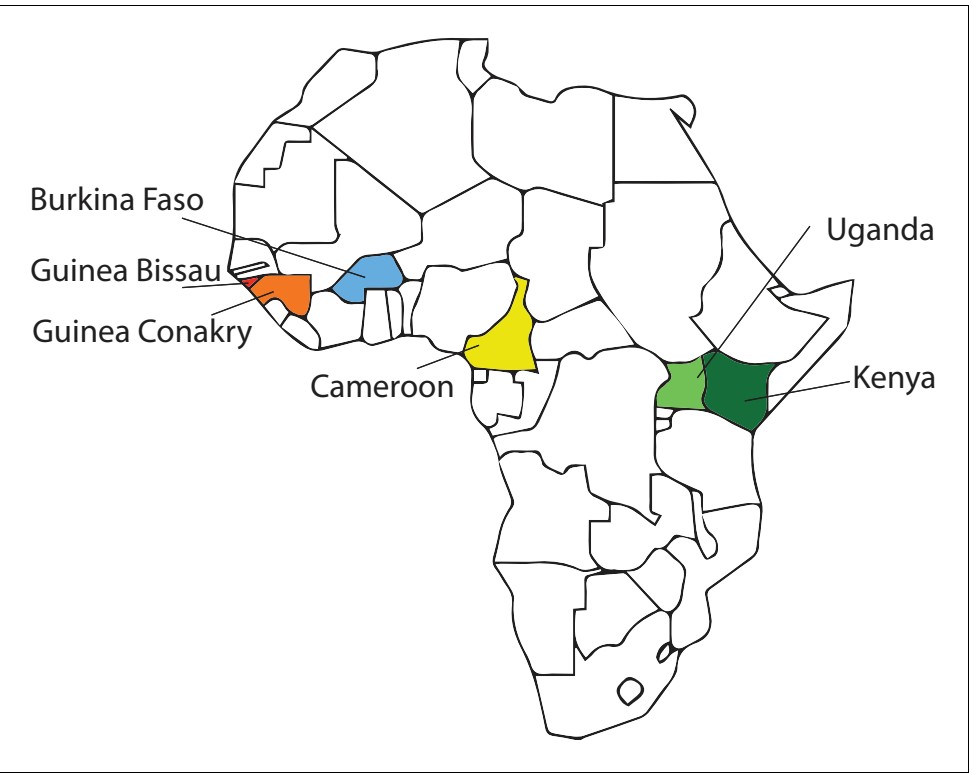

**Figure 1.** African study sites used in study of the 2La inversion. Samples of *Anopheles gambiae* complex from sites in Kenya (dark green), Republic of Guinea (Guinea-Conakry, orange) and Burkina Faso (blue) were used to assess the association between the 2La inversion and susceptibility to *Plasmodium falciparum* infection presented in *Figure 2*. Samples collected from Kenya, Guinea-Conakry, Cameroon (yellow), Uganda (light green), Burkina Faso, and Guinea Bissau (red) were also used to assess the phylogenetic relationship of 2La inversion alleles, presented in *Figure 6*.

The effect of the 2La inversion across taxa was examined in the infection samples containing multiple taxa. The Burkina Faso samples were comprised of *A. gambiae*, *A. coluzzii*, and the Goundry form, while the infection samples from Guinea-Conakry and Kenya were comprised entirely of *A. gambiae*. Thus, infection was analyzed both within and across the taxa of the Burkina Faso samples (*Figure 2—figure supplement 2*). The same association of 2La genotype and infection levels as in the pooled Burkina Faso samples (*Figure 2A*) was observed independently of taxa when analyzed on this fine scale. In all pairwise comparisons, 2L+$^a$/2L+$^a$ homozygotes were significantly more susceptible than 2La/2La homozygotes from any taxon.

All three infection regimes displayed significant association between 2La inversion genotypes and infection (*Figure 2A*). The greater statistical significance of association in the membrane-infected (Burkina Faso) mosquitoes as compared to wild-caught (Kenya and Guinea-Conakry) is probably due in part to greater sample size, but another factor is the method of infectious challenge. In the membrane infections, all mosquitoes were challenged with known *Plasmodium*-gametocytemic blood, and unfed mosquitoes were removed before analysis. Conversely, for the wild-caught mosquitoes, exposure to *Plasmodium* gametocytes or even human blood was unknown, and thus uninfected individuals could result from the genetic effect of the 2La inversion but also from absence of any parasite challenge in previous bloodmeals. Consequently, the wild-caught samples are a more stringent test of association, because multiple irrelevant causes can generate uninfected mosquitoes. The 2La inversion association with infection under the controlled experimental infection regime (Burkina Faso) indicates that differences in intrinsic physiological susceptibility explain at least some of the observed infection difference between mosquitoes carrying alternate inversion forms.

The previous study of the 2La inversion and *P. falciparum* infection, using wild-caught *A. gambiae* in Kenya with cytogenetic karyotyping of 2La and circumsporozoite protein (CSP) immunoassay for

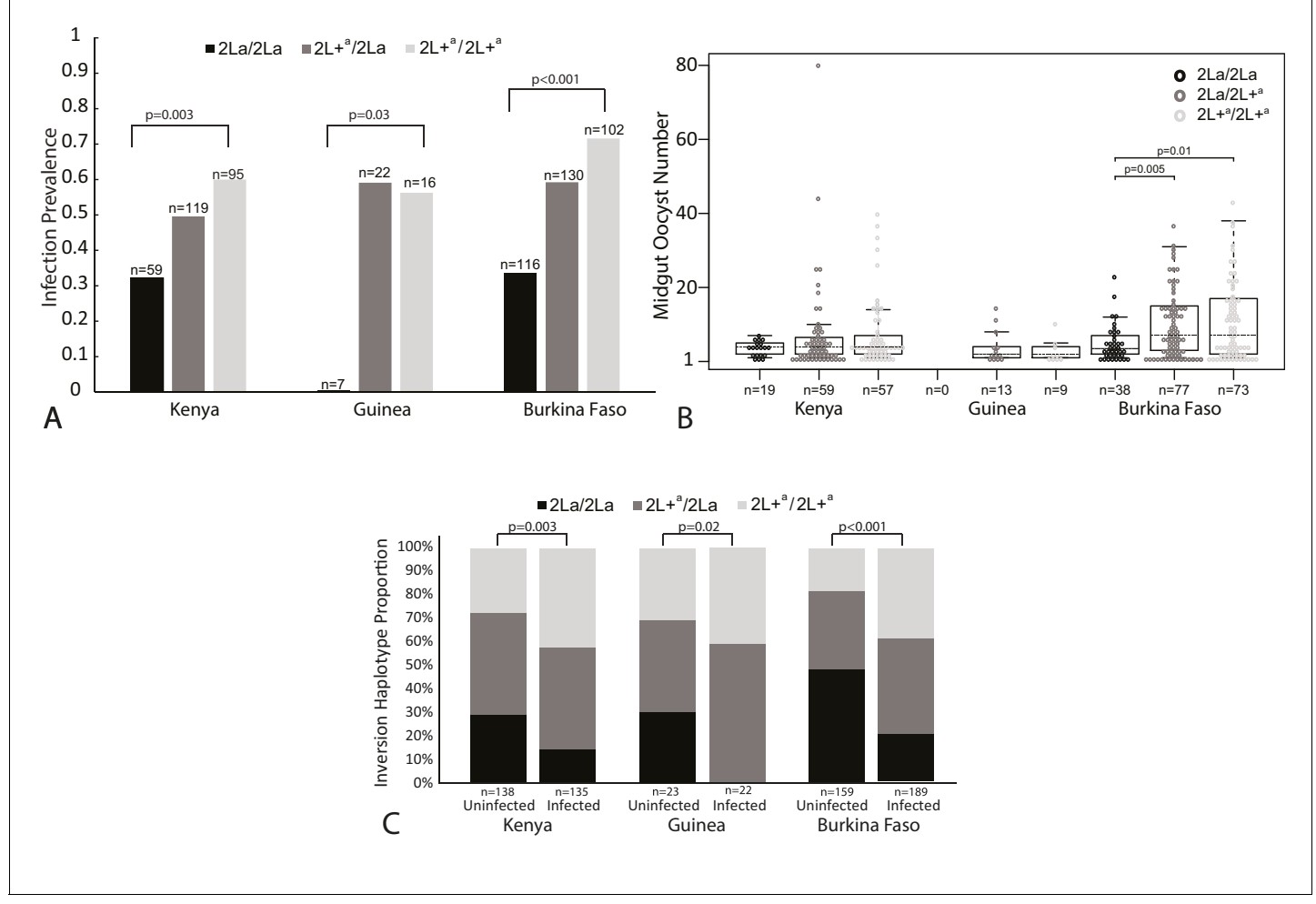

**Figure 2.** *Plasmodium falciparum* infection is associated with the 2La inversion genotype in the *Anopheles gambiae* complex across Africa. (**A**) Oocyst infection prevalence. Sampled mosquitoes were dissected to count midgut oocysts and were molecularly genotyped for the 2La inversion. Differences in oocyst infection prevalence between all three inversion genotypes are indicated by vertical bars, and statistical difference between inversion homozygotes, 2La/2La and 2L+$^a$/2+$^a$, at each geographic sampling site are indicated by p-values. Pairwise statistical comparisons between all inversion genotypes are presented in *Figure 2—figure supplement 1*. Oocyst infections were measured in wild-caught adult mosquitoes after uncontrolled exposure to bloodmeals and parasites in nature (Kenya and Guinea-Conakry), or in adults grown from wild larvae, then membrane-fed on blood from natural gametocyte carriers (Burkina Faso). In Kenya, mosquitoes were held for 7 days before dissection, whereas in Guinea-Conakry, mosquitoes were dissected upon capture. Association was measured in test sets including all infected mosquitoes and an equal number of random non-infected mosquitoes from the same collection, to normalize average oocyst infection prevalence to ≥50% for each site (see Materials and methods). This analysis provides the statistical power to detect both increase and decrease of in prevalence, and normalizes statistical power across all sites. The Burkina Faso collection was comprised of multiple taxa, which are presented and analyzed separately in *Figure 2—figure supplement 2*. (**B**) Oocyst infection intensity. The numbers of midgut oocysts 7–8 days after infection are shown for all individuals with ≥1 midgut oocyst. Statistical analysis by non-parametric Wilcoxon-Mann-Whitney tests indicates that within the Burkina Faso samples, the 2La/2La mosquitoes had significantly lower parasite loads as compared to 2La/2L+$^a$ or 2L+$^a$/2L+$^a$. A similar trend is evident though non-significant in the Kenya samples (all p values shown in *Figure 2—figure supplement 1*). Boxplots delineate the first and third quartiles, median is indicated by the dashed line within the box and error bars are 1.5 time the interquartile range. (**C**) 2La genotype frequency as a function of infection outcome. At each sampling site, 2La/2La homozygotes are more prevalent in the uninfected group, whereas 2L+$^a$/2L+$^a$ homozygotes are more prevalent in the infected group. 2La inversion frequencies are significantly different between the uninfected and infected samples at each of the three sampling sites.

The following figure supplements are available for figure 2:

**Figure supplement 1.** p-Values for all Chi-Square post-hoc comparisons.

**Figure supplement 2.** Taxon breakdown and infection analysis of samples from Burkina Faso.

parasite detection, found that the higher rate of CSP-positives in 2L+$^a$/2L+$^a$ mosquitoes was observed in the mosquito midgut, head-thorax, and salivary glands (*Petrarca and Beier, 1992*). This indicates that the higher prevalence of midgut oocyst infection was translated into sporozoites, and therefore a higher prevalence of infectious 2L+$^a$/2L+$^a$ mosquitoes with positive salivary glands. The current study used molecular genotyping of the 2La inversion with microscopic counting of midgut oocysts to confirm and strengthen the genetic association, to geographically extend the effect to West as well as East Africa, and to indicate that the effect is at least partly due to genetically controlled physiological difference in susceptibility. However, in addition to physiological susceptibility, to which we return in the Discussion, other factors could contribute to a 2La-associated mechanism influencing malaria transmission and epidemiology. These factors could potentially include variation for mosquito longevity, general immune competence, and behavior, and here we examine each of these possible components of the mechanism.

## The 2La inversion is not associated with mosquito longevity in semi-field conditions

A possible explanation for the greater *P. falciparum* infection levels of 2L+$^a$/2L+$^a$ mosquitoes could be that these individuals display a low overall level of immune competence. In other words, perhaps they are highly infected with malaria not as a specific effect, but because they have poor antimicrobial defenses against all infections. Moreover, if the more infected 2L+$^a$/2L+$^a$ mosquitoes die sooner than 2La/2La individuals, regardless of the cause, then the epidemiological impact of the 2La inversion effect on susceptibility could be diminished, because infected mosquitoes become infectious for malaria transmission only when sporozoites invade the salivary glands 12–14 days after the infective bloodmeal.

To test these two hypotheses, we compared longevity of 2La genotypes in a natural Kenyan *A. gambiae* population segregating all three 2La genotypes. Mosquitoes grown from wild-collected larvae were maintained in open outdoor screen cages under semi-field conditions, exposed to natural environmental fluctuation, and environmental pathogens and microbes, and survivorship was measured. There was no significant difference in mosquito life span between 2La/2La and 2L+$^a$/2L+$^a$ mosquito genotypes in the sympatric population across three replicate experiments (*Figure 3*, p = 0.478, average median survival across replicates 19.3 days). This result also suggests that the greater *P. falciparum* infection levels of 2L+$^a$ carriers is not due to a general low immune competence to all pathogens, or else the 2L+$^a$/2L+$^a$ mosquitoes should have displayed lower survival under these conditions. It is expected that under fully wild conditions, the different inversion genotypes would physically migrate to preferred ecological niches, and thus holding all genotypes under uniform conditions as done here probably represents a more stringent test for difference in survivorship.

## The 2La inversion is not associated with difference in overall immune competence

The lack of difference for longevity when exposed to environmental microbes and pathogens suggests that the higher *P. falciparum* infection rates of 2L+$^a$/2L+$^a$ mosquitoes are probably not a consequence of general immune deficiency. However, we could not be certain that the ambient natural microbes in the longevity test (*Figure 3*) included highly virulent mosquito pathogens. Therefore, we employed a known virulent pathogen, the entomopathogenic fungus, *Metarhizium*, in a direct assay for difference in functional immunity between 2La inversion genotypes. Entomopathogenic fungi are model eukaryotic pathogens used in studies of mosquito immunity and are a proposed biopesticide for vector control (*Dong and Dimopoulos, 2009*; *Thomas and Read, 2007*). We reasoned that if the 2L+$^a$ -associated susceptibility to malaria is a non-specific effect of lower general immune competence, then patterns of 2La genotype association similar to those observed with malaria infection should be observed after fungal infection. Conversely, absence of 2La genotype association for fungus susceptibility would indicate that the *P. falciparum* effect must be due to a more specific mechanism of susceptibility.

Similar to the longevity test, the mosquitoes were adults emerged from wild-caught larvae from a Kenyan population segregating all three 2La genotypes. The virulence of *Metarhizium* as a pathogen was confirmed by the shortened mosquito lifespan observed across all 2La inversion genotypes

(*Figure 4*, average median survival across replicates 6 days as compared to 19.3 days without fungus exposure in *Figure 3*), but there was no difference in survival between 2La genotypes following infection with entomopathogenic fungus (*Figure 4*, p=0.877). The survival results taken together indicate that 2La inversion genotypes do not differ in longevity or overall immune competence, and that the mechanism of *P. falciparum* susceptibility associated with the 2L+$^a$ inversion allele is more specific than, and cannot be explained by, generalized immune incompetence.

## The 2La inversion is not associated with differential rates of human blood feeding

The probability of vector infection with *P. falciparum* is directly dependent upon mosquito propensity for human blood feeding. The prior report on 2La and infection did not detect a difference between 2La karyotypes in Kenya for carriage of human blood, measured by immunoassay in engorged samples, or in the host-seeking behavior by response to human versus cow odor-baited traps (*Petrarca and Beier, 1992*). Here, we determined bloodmeal sources using a molecular blood assay in wild-caught *A. gambiae* adult mosquitoes carrying visibly detectable blood in Guinea-Conakry (indoor-resting aspirator capture n = 126, human landing capture indoors n = 72, and human landing capture outdoors n = 12). Across all 2La inversion genotypes, 100% of bloodmeals from the blood-containing females were human-derived (total n = 210, 2L+$^a$/2L+$^a$ n = 69, 2L+$^a$/2La n = 72, 2La/2La n = 69). Thus, there was no difference in bloodmeal carriage across 2La inversion forms, which confirms in West Africa the earlier data from Kenya. These results indicate that the reason for greater natural infection levels of the 2L+$^a$ homozygotes is not due to a differential human feeding rate.

## The 2La inversion may be associated with behavioral differences

The association of 2La genotype with host-seeking behavior was also analyzed directly using human landing capture (HLC) to sample human-seeking *A. gambiae* mosquitoes at sites in Guinea-Conakry segregating all three 2La genotypes (Koundara and Koraboh). Mosquito samples collected inside of houses (indoor HLC) and outside ≥10 m from a dwelling (outdoor HLC) indicate endophilic and exophilic tendencies, respectively. Site and time-matched larval collections were also sampled from larval pools present in the villages. Larval pools are a limiting resource for female oviposition, and collection of larvae thus provides an unbiased measure of overall population genotype frequencies, independent of adult mosquito behavior or resting site preference (*Fillinger and Lindsay, 2011*; *Gordicho et al., 2014*; *Riehle et al., 2011*; *Killeen et al., 2002*). Unbiased sampling of adult mosquitoes is not currently possible by other methods.

Examination of the Guinea-Conakry population sampling data reveals two interesting behavioral effects (*Figure 5*). First, 2L+$^a$/2L+$^a$ mosquitoes are less likely than 2La/2La genotypes to be captured indoors attracted to human bait, suggesting that 2L+$^a$/2L+$^a$ mosquitoes are less likely than 2La/2La to feed inside habitations. The deficit of 2L+$^a$/2L+$^a$ homozygotes among indoor captured mosquitoes was significant (double asterisks in *Figure 5*, Chi-squared = 6.99, df = 1, p=0.008, for the fraction of 2L+$^a$/2L+$^a$ caught on HLC indoor versus caught on all other methods, as compared to the same fraction for 2La/2La). Second, the reduced indoor capture rate of 2L+$^a$/2L+$^a$ mosquitoes was not compensated by elevated outdoor HLC collection of this genotype. Thus, proportionally fewer 2L+$^a$/2L+$^a$ mosquitoes were captured by HLC overall (HLC outdoor and indoor combined), relative to their total representation in the population based on larval site frequencies (2L+$^a$/2L+$^a$ were 27% of total HLC versus 42% of total larvae). This under-representation in adult population sampling of 2L+$^a$/2L+$^a$ genotypes as compared to 2La/2La was also significant (Chi-squared = 5.92, df = 1, p=0.02). In contrast, the inverse pattern was observed for 2La/2La mosquitoes, which represented 28% of total larvae but 40% of total HLC. Heterozygotes represented the balance of the sample, about one third of larvae and of total HLC. These two effects, a lower rate of indoor host-seeking for 2L+$^a$/2L+$^a$ relative to 2La/2La, and apparent under-sampling of 2L+$^a$/2L+$^a$ adults relative to their overall population frequency, are also observed as a consistent tendency at each of the two individual sites (*Figure 5—figure supplement 1*). The sample sizes are smaller when sites are considered individually, but the effect is still significant at Koundara alone (Chi-squared = 4.59, df = 1 p=0.03), and for both sites when individual p-values are combined by the method of Fisher (combined p=0.02, Koundara p=0.03, Koraboh p=0.08).

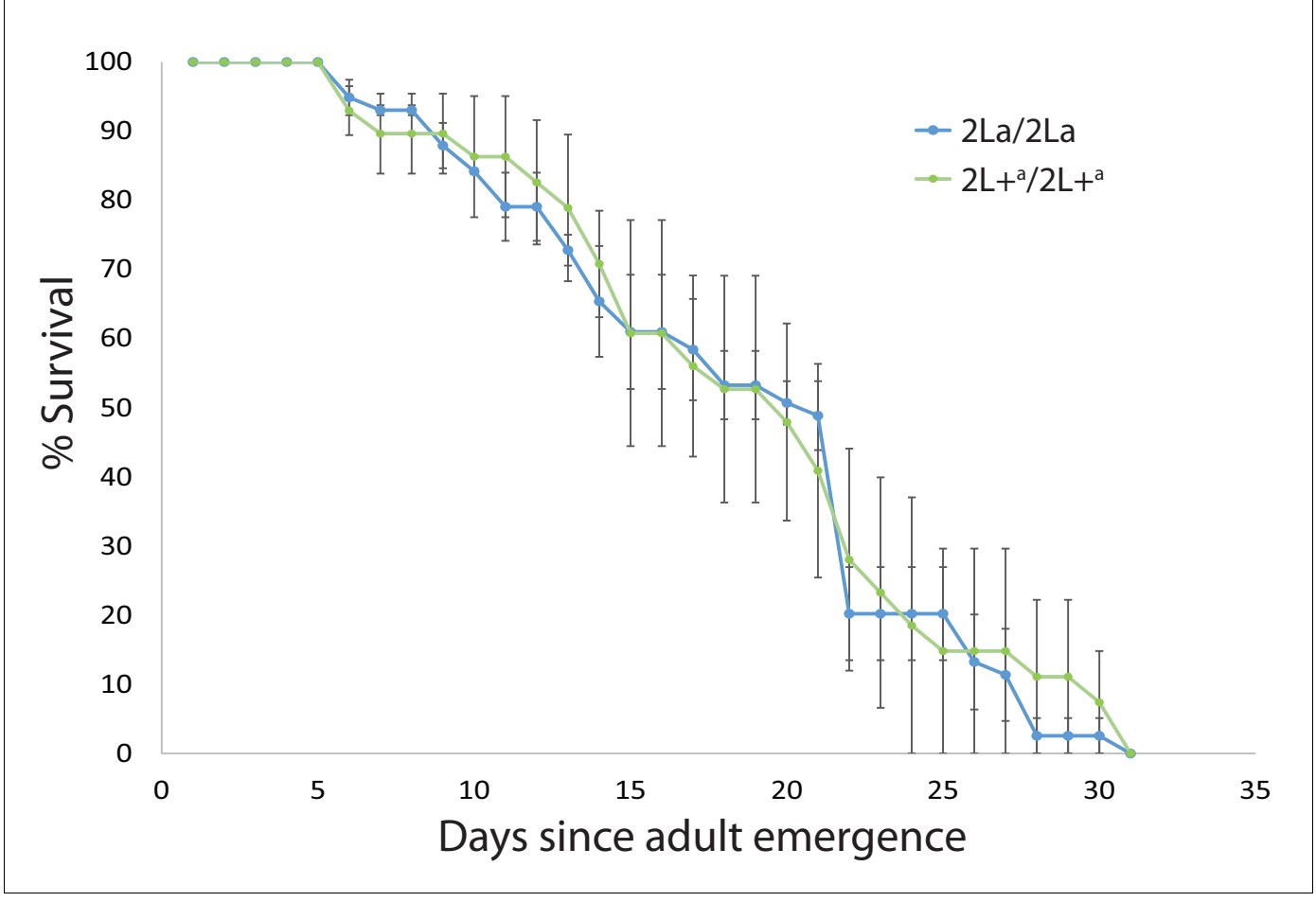

**Figure 3.** 2La inversion homozygotes display equivalent longevity under semi-field conditions. Wild *A. gambiae* larvae were collected in western Kenya, and the emerged adults were maintained in outdoor cages, exposed to ambient conditions including environmental microbes and pathogens. Dead mosquitoes were collected daily and genotyped for the 2La inversion. Plot indicates average longevity with standard error for 2La/2La and 2L+[a]/2L+[a] homozygotes in three replicate experiments (median longevity 19.3 d). The absence of survival difference between 2La inversion homozygotes exposed to natural microbes and pathogens suggests that the higher malaria infection levels of 2L+[a]/2L+[a] mosquitoes is not a consequence of general immune deficiency.

## The 2La inversion has a monophyletic origin

We examined the evolutionary origin of the 2La inversion using *A. gambiae* population resequencing data available for the first time from across the African continent, including the Goundry form from Burkina Faso. Based on previous manual sequence analysis near the breakpoints, the inversion has been regarded as either polyphyletic (*Caccone et al., 1998*) or more recently monophyletic (*Sharakhov et al., 2006*). The history of introgression of the 2La inversion throughout the *A. gambiae* complex is subject to some uncertainty (*O'Loughlin et al., 2014*; *Sharakhov et al., 2006*). Using complete genome data, we were now able to examine patterns of similarity between alternate inversion forms within and outside the boundaries of the 2La inversion.

Six genomic windows of 5 megabases (Mb) each were analyzed for phylogenetic relatedness in *A. gambiae* samples from diverse geographic populations across Africa. If the inversion has a monophyletic origin, the phylogenetic clustering of samples should be driven by 2La genotype status at inversion breakpoints and within the inversion, regardless of the geographic source, while sequences outside of the inversion should cluster by geographic origin of the samples. All windows displayed phylogenetic patterns consistent with monophyly (*Figure 6*). For the two windows spanning the inversion breakpoints and the one window central to the inversion, all individuals cluster predominantly by 2La genotype regardless of their geographic origin, despite apparent geographic

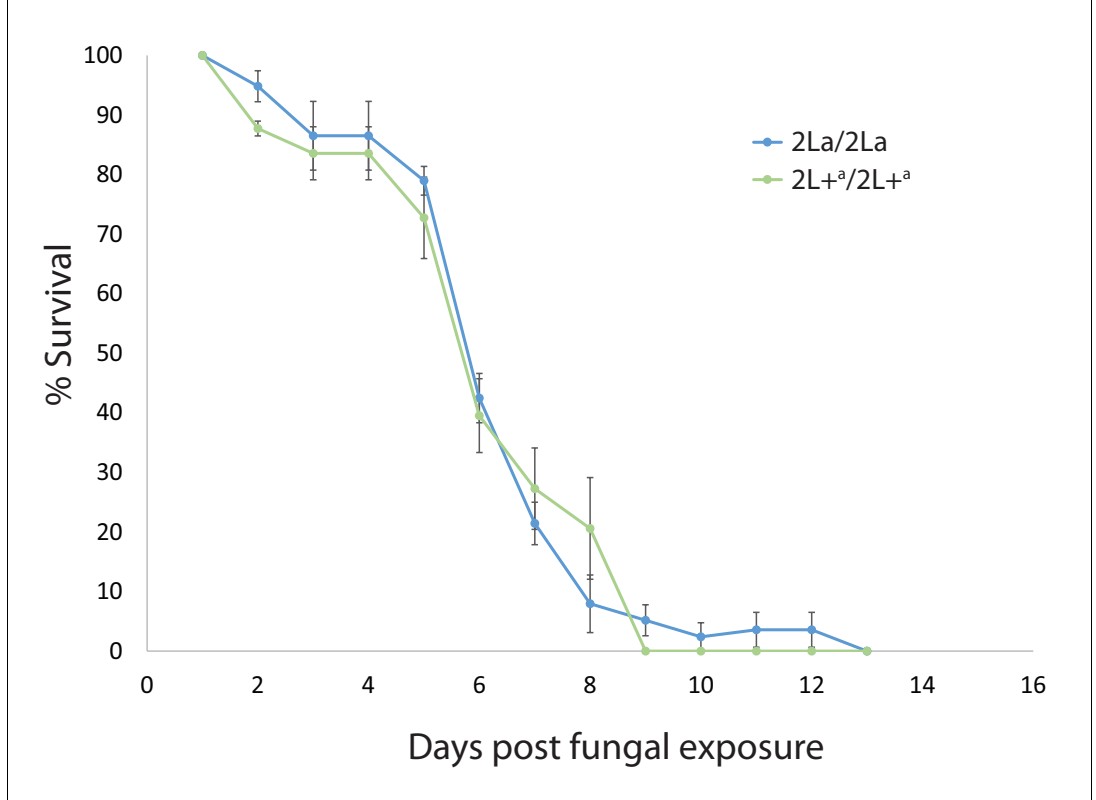

**Figure 4.** 2La inversion homozygotes are equally susceptible to pathogenic fungus. As a direct test of general immune competence, adult *A. gambiae* grown from wild larvae in western Kenya were exposed to spores of the insect fungal pathogen, *Metarhizium anisopliae* strain ICIPE30. Fungus-exposed mosquitoes were maintained in outdoor cages and 2La genotypes were assayed for longevity. Plot indicates average longevity with standard error for 2La/2La and 2L+ᵃ/2L+ᵃ homozygotes in three replicate experiments. Fungal exposure decreases longevity as compared to non-exposed controls (median longevity 6 days, compare with 19.3 days in *Figure 3*), but rates of fungal killing were not different between inversion homozygotes. The 2L+ᵃ/2L+ᵃ mosquitoes are immune competent against *Metarhizium* but significantly more susceptible to malaria infection, indicating a relatively specific mechanism of susceptibility to malaria, rather than nonspecific low immune competence.

substructure visible in some samples. In contrast, the control windows (one outside the inversion on the left arm of chromosome 2L, and one on each of the arms of chromosome 3) displayed a pattern of clustering based mostly on geographic origin of the samples. Longer branch lengths observed for some samples are likely due to inbreeding, as described for the Goundry form (*Crawford et al., 2016*) and Kenya (*Miles et al., 2016*) source populations.

Thus, 2La inversion alleles share a common evolutionary history within the *A. gambiae* complex that is deeper than geography or taxonomic group. This observation is consistent with the finding of a shared genetic association with malaria infection across geography and taxa, and strongly suggests that the infection phenotype is controlled by a common genetic mechanism carried and spread by the 2La inversion.

## 2La inversion distribution correlates with annual precipitation

We analyzed the spatial distribution of 2La inversion genotypes on a fine geographic scale, using empirical data we collected across a ~350 km ecological transect in Guinea-Conakry and southern Mali (*Coulibaly et al., 2016*), and annual precipitation data (*Hijmans et al., 2005*). There is a strong positive correlation between the frequency of the 2L+ᵃ allele and annual precipitation (*Figure 7*, $r^2 = 0.86$). The genotype data included two taxa, *A. gambiae* and *A. coluzzii*, and examination of the taxa individually, although generating smaller sample sizes, nevertheless still demonstrated a strong positive correlation between 2L+ᵃ allele and annual precipitation (*Figure 7—figure supplement 1 A. gambiae* $r^2 = 0.857$, *A. coluzzii* $r^2 = 0.999$).

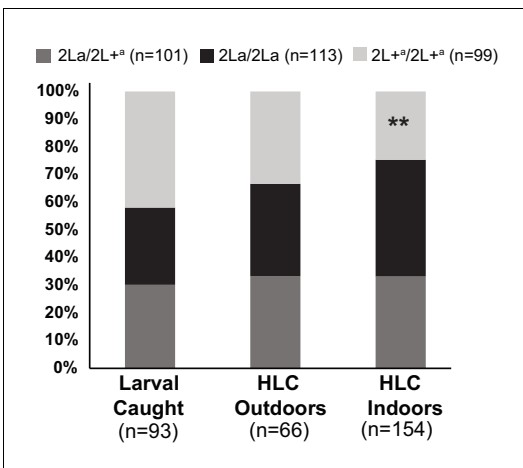

**Figure 5.** Individuals carrying the 2L+$^a$ inversion allele are less likely to be captured indoors. Adult mosquitoes were sampled in Koraboh and Koundara, Guinea-Conakry by human landing capture (HLC) inside houses or outdoors (≥10 m from the nearest house), and larvae were also collected in the same villages. Of the 2L+$^a$/2L+$^a$ homozygotes captured in both villages, a significantly smaller fraction was captured by indoor HLC as compared to the fraction of 2La/2La homozygotes captured by indoor HLC (p=0.008, double asterisk). In addition, there was a significant deficit of 2L+$^a$/2L+$^a$ among all captured adults (indoor HLC + outdoor HLC) relative to the expectation from their overall population representation in larval site frequencies, as compared to 2La/2La adults relative to their larval frequencies (p=0.02). The same tendencies are observed for the two individual sites (*Figure 5— figure supplement 1*).

The following figure supplement is available for figure 5:

**Figure supplement 1.** Spatial partitioning of 2L+$^a$ carriers is reproduced at individual Guinea-Conakry study sites.

## Discussion

### Population genetic context of 2La inversion-associated susceptibility

Here, we report the significant association of 2La inversion alleles with *P. falciparum* infection in *A. gambiae*-taxa vector populations across Africa. Most previous genetic studies of *A. gambiae* susceptibility to *P. falciparum* were laboratory studies, and the traits were not tested in the natural population. Several previous genetic association studies of susceptibility were done in the natural vector population, but the variants were not geographically replicated, and the frequencies of the traits in nature were not determined (*Harris et al., 2010*; *Horton et al., 2010*; *Luckhart et al., 2003*; *Riehle et al., 2006*; *Mitri et al., 2015b*). Thus, based on results of the current study, the 2La inversion is, to our knowledge, the only frequent and widespread natural genetic polymorphism with significant influence on *P. falciparum* infection rates in the vector population.

The association of malaria infection and 2La inversion genotype is probably due to a common underlying genetic mechanism throughout Africa rather than a series of local adaptations, because of the monophyletic origin of the 2La inversion within the *A. gambiae* species complex, and the Africa-wide distribution of the allele-phenotype association. The inversion genotype was significantly associated with oocyst infection prevalence at all three sites tested, and with infection intensity at the Burkina Faso site. Oocyst prevalence is the most relevant parameter for malaria transmission, because it distinguishes between mosquitoes competent for transmission from those that are not. In nature most mosquitoes carry <5 oocysts (*Pringle, 1966*).

The mechanism of differential infection of 2La inversion genotypes is at least partly based on genetically controlled physiological differences for parasite development efficiency in 2L+$^a$/2L+$^a$ as compared to 2La/2La mosquitoes. The molecular mechanism of differential susceptibility must be relatively specifically addressed to *P. falciparum*, because the 2La genotypes do not display detectable differences in general immune competence. The pathogen spectrum of the 2La susceptibility trait, if it extends to other microbes beyond malaria, is unknown. However, attempts to identify the underlying causative genetic mutations or causative mechanism distinguishing the 2La and 2L+$^a$ alleles are not likely to be fruitful, because the mechanism is probably an additive or synergistic effect among variants dispersed throughout the 22 Mb inversion. The inversion alleles are non-recombining haplotypes carrying ~2000 genes. SNPs within many of these genes, both the few relevant genes and the large mass of irrelevant ones, are probably equally linked to inversion-associated phenotypes (*Corbett-Detig and Hartl, 2012*).

Despite the challenges of genetic comparison between inversion alleles with suppressed recombination, standard genetic approaches can be applied in crosses of inversion homozygotes, in which chromosomes are free to recombine. Genetic mapping carried out in wild pedigrees from homozygous 2La/2La crosses in Mali identified a cluster of quantitative trait loci (QTL) linked to oocyst

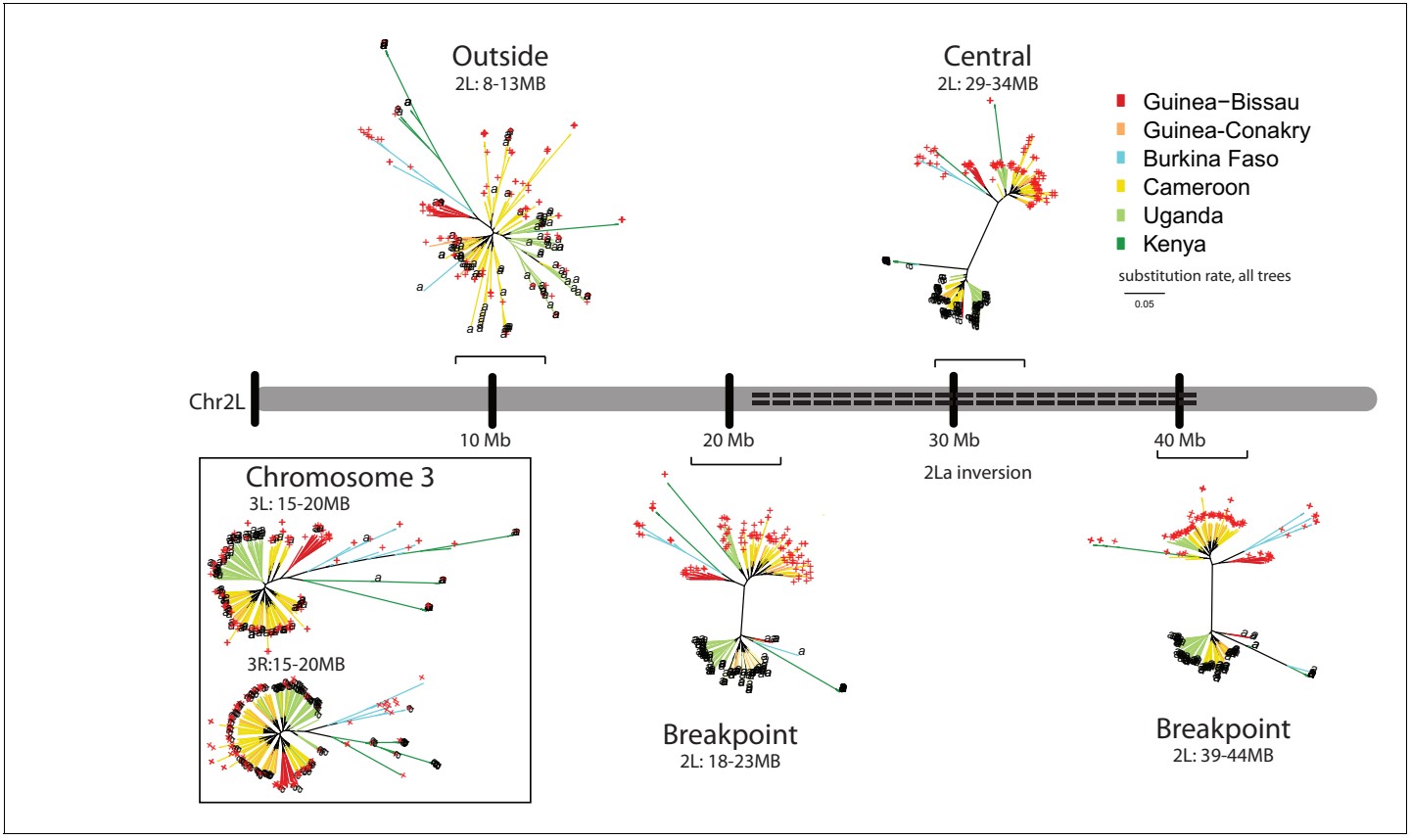

**Figure 6.** The 2La inversion has a monophyletic origin throughout Africa. Phylogenetic trees were constructed for genomic windows of 5 megabases (Mb) each extracted from whole-genome sequences of 103 2La/2La and 107 2L+$^a$/2L+$^a$ wild *A. gambiae* mosquitoes, obtained with permission from the *Anopheles gambiae* 1000 (Ag1000) Genomes Consortium (*Miles et al., 2016*), and whole-genome sequences of 1 2La/2La and 11 2L+$^a$/2L+$^a$ wild Burkina Faso *A. gambiae* Goundry form mosquitoes (*Crawford et al., 2016*). Branch color indicates the country origin of the sample (see country key). The terminal label at the end of each branch indicates genotype of the 2La inversion (black 'a' for 2La/2La, and red '+' for 2L+$^a$/2L+$^a$). The 2La inversion genotypes of Ag1000 samples were determined informatically (*Figure 6—figure supplement 1*), and Goundry form samples were genotyped by molecular diagnostic assay. The chromosome map depicts the left arm of chromosome 2 (Chr2L), with the centromere to the left, and the dashed line indicates the extent of the 2La inversion. Brackets indicate the genomic windows at three distinct positions relative to the 2La inversion: (i) Spanning the proximal and distal inversion breakpoints (respectively, 2L:18–23 Mb and 2L:39–44 Mb), (ii) in the central region of the inversion (2L: 29–34 Mb), and controls outside the 2La inversion on the same chromosome (2L:8–13 Mb) or on Chromosome 3 (3L:15–20 Mb and 3L:15–20 Mb). Breakpoint and inversion-central trees display a subdivision between 2La inversion genotypes, regardless of the geographic origin of the samples, while trees outside the inversion cluster by geography and not inversion genotype. Branch length scale under the key indicates nucleotide substitutions per site and applies to all six trees.

The following figure supplement is available for figure 6:

**Figure supplement 1.** Principal components analysis (PCA) for 2La genotype assignment of whole-genome sequenced samples.

infection prevalence (*Niaré et al., 2002*; *Riehle et al., 2006*). The cluster is located within the proximal 2La inversion breakpoint, and the region was termed the *Plasmodium* Resistance Island. Within the resistance island, the APL1 gene family of leucine-rich repeat (LRR) proteins display a protective phenotype for *Plasmodium* infection prevalence in functional assays (*Mitri et al., 2009*; *Riehle et al., 2008*). Another LRR immune factor, LRIM1, is also located within the 2La inversion, and additional nearby LRR genes with a similar protective phenotype for *P. falciparum* were recently identified (*Mitri et al., 2015a*). However, to date, there is no evidence that genetic variation at these candidate genes is associated with the observed phenotypic differences between 2La and 2L+$^a$ inversion alleles, and identification of such variation between inversion alleles may remain elusive for the reasons enumerated above.

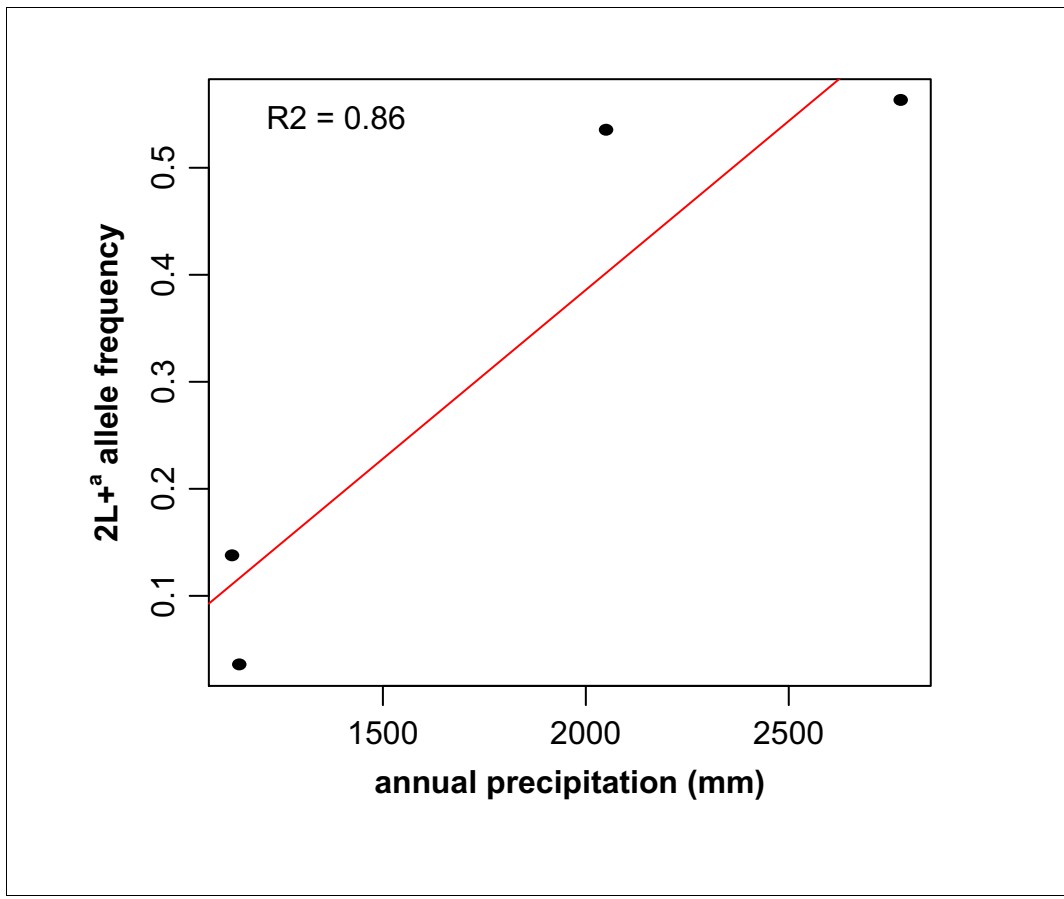

**Figure 7.** The frequency of the 2La inversion is correlated with ecology. Frequency of the 2L+[a] allele was plotted as a function of annual rainfall across an ~350 km ecological transect from arid savanna in southern Mali to deep forest in southern Guinea-Conakry, and displayed a positive correlation ($r^2$ = 0.86, sample sizes from lowest to highest 2L+[a] frequency, respectively, are n = 190, 191, 365, and 259). Because the samples used in this analysis included individuals from two taxa, *A. gambiae* and *A. coluzzii*, an additional analysis of ecological correlation was conducted for each taxon, shown in *Figure 7—figure supplement 1*.

The following figure supplement is available for figure 7:

**Figure supplement 1.** Correlation of 2L+[a] frequency with annual rainfall is reproduced in individual mosquito taxa.

If *P. falciparum* infection is deleterious to the vector, the elevated infection prevalence of 2L+[a] carriers might seem inconsistent with the locally high 2L+[a] allele frequencies. However, evidence is lacking that *Plasmodium* imposes a fitness cost upon infected mosquitoes in nature, and deleterious effects of *Plasmodium* are mainly seen using artificial vector-parasite combinations or unnatural infection levels (*Ahmed and Hurd, 2006*; *Ferguson and Read, 2002*; *Robert et al., 1990*; *Zhao et al., 2012*). Parasite reproductive fitness should be maximized by maintaining active, flying vectors that can successfully transmit sporozoites. Alternatively, if natural malaria infection imposes a fitness cost, then the 2L+[a] chromosome presumably would carry linked compensatory mutations for tolerance to higher levels of parasites to mitigate such cost.

## Behavioral tendencies of 2La inversion genotypes

The most likely explanation for the observed behavioral results is that 2L+[a]/2L+[a] mosquitoes are significantly more exophilic, and 2La/2La more endophilic. Exophilic mosquitoes tend to rest outdoors to digest the bloodmeal and complete their gonotrophic cycle, although they may bite either outdoors or indoors, while endophilic mosquitoes both bite and rest indoors (*Bockarie et al., 1993*;

*Coluzzi et al., 1977*; *Gillies, 1956*; *Riehle et al., 2011*). Endophilic mosquitoes are efficiently sampled by capture of house-resting mosquitoes within the enclosed space using pyrethroid spray or aspirator, while sampling methods for exophilic mosquitoes are inefficient because the outdoor resting sites are spatially scattered at largely uncharacterized sites anywhere in the outdoor environment (*Gillies, 1956*, *1954*). Therefore, we presume that the 2L+$^a$/2L+$^a$ mosquitoes not collected in the peridomestic environment by either indoor or outdoor HLC were located in more distant and dispersed outdoor resting sites. An alternate hypothesis could be that 2L+$^a$/2L+$^a$ larvae simply fail to survive to the adult stage with the same success as 2La/2La carriers. This argument is unconvincing for several reasons, including (i) the above data that 2L+$^a$/2L+$^a$ mosquitoes do not display lower survival, and are not immunologically weaker than 2La/2La, (ii) evidence that females preferentially choose oviposition sites that maximize survival of larval progeny (*Diabaté et al., 2008*; *Munga et al., 2006*; *Suh et al., 2016*), and (iii) if 2L+$^a$/2L+$^a$ larvae consistently displayed elevated mortality in nature, it would be difficult to explain the observed frequencies of a genotype with high negative fitness load.

The feeding behavior of the presumably exophilic fraction of 2L+$^a$/2L+$^a$ mosquitoes that are not captured as adults by HLC cannot be determined directly, and it cannot be ruled out that they could be feeding on animals and resting outdoors. However, among the large fraction of capturable 2L+$^a$/2L+$^a$ mosquitoes, the rate of human bloodmeal carriage was 100%, and the level of *P. falciparum* infection was significantly elevated as compared to 2La/2La. It is not likely that the non-captured 2L+$^a$/2L+$^a$ mosquitoes would display a radically different behavior (i.e. zoophilic) or biology than the capturable ones, for several reasons. First, most knowledge of *A. gambiae* behavior to date indicates strong human-feeding propensity (*Costantini and Diallo, 2001*; *Garrett-Jones et al., 1980*; *Gillies and Coetzee, 1987*; *White, 1974*; *Pates et al., 2001*), with very few accounts of zoophily (*Diatta et al., 1998*; *Duchemin et al., 2001*), so an exclusively zoophilic group within *A. gambiae* would be unexpected. Second, the only previous test of association of the 2La inversion with endophilic or exophilic behavior, in northern Nigeria, found that 2L+$^a$ carriers were less likely than 2La carriers to be captured indoors (*Coluzzi et al., 1979*), which is consistent with the current results. Third, carriage of the 2L+$^a$/2L+$^a$ genotype is a hallmark of the sylvan-adapted Forest chromosomal form (*Coluzzi et al., 1979*), which was observed to be highly exophilic and human-biting (*Bockarie et al., 1993*, *1994*).

Taking together our results and the literature, the non-captured 2L+$^a$/2L+$^a$ mosquitoes in our study most likely indicate a behavioral tendency toward exophily of anthropophilic 2L+$^a$/2L+$^a$ mosquitoes. In other words, the non-captured fraction represent one tail of an overall 2L+$^a$/2L+$^a$ distribution that is skewed toward exophily, while 2La/2La behavior is skewed toward endophily, and both are probably equally anthropophilic. It is this behavioral tendency of 2L+$^a$/2L+$^a$ mosquitoes that makes them a potential concern, because under the pressure of indoor-based vector control such as ITNs and IRS, the relative population frequencies of the 2L+$^a$ allele would be expected to increase, and the resulting population could become more exophilic. In the only examination of 2La genotypes under indoor selection pressure to our knowledge, the 2L+$^a$/2L+$^a$ frequency increased from 0% detectable in 1994 before ITN rollout in western Kenya to 66% by 2011 after rollout (*Matoke-Muhia et al., 2016*). The 2La inversion is not genetically associated with resistance to the pyrethroid insecticides used on ITNs (*Brooke et al., 2002*), so the underlying mechanism is more likely to be behavioral avoidance rather than chemical selection. At the time of the current collections in Guinea-Conakry, the country had the lowest use of ITNs in Africa (*President's Malaria Initiative, 2014*), which provides the opportunity to monitor 2L+$^a$ allele frequencies there as ITNs are rolled out.

## Epidemiological significance

Here, we demonstrate consistently higher *P. falciparum* infection rates of *A. gambiae* complex 2L+$^a$ inversion carriers across the African continent. Consequently, it might be expected that the frequency of 2L+$^a$ in local vector populations would correlate with human malaria incidence. Targeted studies to correlate the 2La inversion and the prevalence of human *P. falciparum* infection (referred to as *P. falciparum* parasite rate, PfPR) will be necessary to separate multiple potentially confounding factors, and here we discuss the main considerations.

In the absence of consistent public health data from many malaria-endemic African countries, cartographic approaches are used to impute the spatial distribution and epidemiological parameters of

PfPR (*Bhatt et al., 2015*; *Hay et al., 2008*). Examination of ecological correlates found that proxies for humidity (high annual rainfall and vegetation density) were the covariates most strongly correlated with PfPR (*Weiss et al., 2015*). The 2L+$^a$ allele generally correlates geographically with humidity (*Coluzzi et al., 1979*; *Cheng et al., 2012*). Thus, in a broad sense, the ecological conditions associated with the highest PfPR as well as vector 2L+$^a$ frequency are positively correlated, but detailed ecological genetic studies are needed.

Here, we demonstrate a significant positive correlation between annual precipitation and the proportion of the 2L+$^a$ inversion along an ecological transect in Guinea-Conakry and Southern Mali. Epidemiological data do not yet exist to analyze correlation of PfPR and 2La inversion genotype directly because health data in Guinea-Conakry are 'either inaccurate or sparse' (*Tiffany et al., 2016*). However, the highest PfPR in Guinea-Conakry is in the southern forested zone, where about 85–90% of the 5–14 year-old group was *P. falciparum*-positive during the period 2011–2013 (*Tiffany et al., 2016*), near the current study sites of Koraboh and Koundara where vector populations displayed the highest 2L+$^a$ frequencies of up to 57%, while in the northern humid savanna the 2L+$^a$ frequency decreased to 27% and in the arid savanna, 4% (*Coulibaly et al., 2016*).

Thus, the core habitats of 2L+$^a$ carrying mosquitoes appear to be ecologically synonymous with the foci or hotspots of the most stable, endemic core transmission of African *P. falciparum*. It is perhaps not surprising that high PfPR is correlated with availability of water and humidity, conditions that promote robust vector populations. This could suggest that the presence of high frequencies of 2L+$^a$ carrying mosquitoes directly contributes to stable malaria transmission, but distinguishing between correlation and causation will require detailed village-based epidemiological surveys that incorporate 2La inversion genotyping and measurement of ecological parameters.

We speculate that in high-PfPR foci, the epidemiological influence of high 2L+$^a$ frequencies could be partially masked by current high levels of transmission mediated by all vector genotypes together. Under such favorable ecological conditions for both vectors and PfPR, transmission levels may be saturating, and variation in vectorial capacity may not be the most important limiting factor for PfPR (*Brady et al., 2016*). Under the current conditions, human short-duration immunity maintained by frequent exposure to infectious bites may contribute more strongly to defining the upper limit of PfPR (*Langhorne et al., 2008*; *Smith et al., 2006*; *Tran et al., 2013*).

The current regions of high PfPR, 2L+$^a$ frequency, and humidity are also some of the economically poorest and geographically remote, where malaria control has been less consistently implemented (*Tiffany et al., 2016*; *Bhatt et al., 2015*). However, as sustained malaria control interventions at these high-PfPR foci depress vector populations and baseline PfPR, the more susceptible and exophilic 2L+$^a$ vectors may begin to disproportionately mediate residual transmission. The numbers of 2L+$^a$/2L+$^a$ in the population, as well as their contribution to the entomological inoculation rate (EIR), are likely to be systematically underestimated if, as suggested above, they are not efficiently sampled by standard adult collection methods. The propensity of 2La/2La to rest indoors and be oversampled may cause overestimation of the impact of ITNs and IRS, while the exophilic behavior and undersampling of 2L+$^a$/2L+$^a$ may generate a partially invisible reservoir of efficient transmission.

Collection of additional empirical data from multiple sites will be required for accurate modeling of the contribution of 2La inversion genotypes to transmission during ITN and IRS rollout. The necessary data include *P. falciparum* infection prevalence per 2La inversion genotype, seasonal inversion genotype frequency, and the force of negative and positive selection by ITNs and IRS upon the different inversion genotypes. Another relevant variable is declining human protective immunity as the parasite challenge rate decreases. The key question is whether, in the humid high-PfPR and high-2L+$^a$ core ecological niches of *P. falciparum* transmission, existing tools could sufficiently depress the parasite reproductive rate in residual transmission hotspots (*Bousema et al., 2012*). Or, alternatively, whether an exophilic vector reservoir would require additional or novel vector control solutions.

## Conclusions

The humid equatorial zones in Africa are core ecological niches of the malaria transmission system, where transmission is particularly well adapted to the environment and should be most stably implanted. High frequencies of the malaria-susceptible 2L+$^a$ vector type is a correlated feature of these humid, high-PfPR zones. The specific contribution of 2L+$^a$ carriers to the ecological durability of malaria transmission may currently be obscured by conditions of high general PfPR in 2L+$^a$-rich zones, combined with relatively low coverage of these areas by large-scale malaria control efforts, as

well as sparse research and health statistics. As PfPR recedes in these sites with the expansion of large-scale malaria control measures, the influence of high frequencies of 2L+[a] could become more apparent, and pose new challenges. The presence of abundant 2L+[a] carriers may require different methods to efficiently sample and measure exophilic adult mosquitoes, coupled with new approaches for their control.

## Materials and methods

### Mosquito collections for *P. falciparum* infection levels

Mosquitoes collected at multiple African sites were comprised of taxa of the *A. gambiae* species complex. For all mosquitoes reported in the current work and wherever not specifically stated, the taxa were: Kenya, *A. gambiae* is the only complex member in East Africa; Guinea-Conakry, *A. gambiae* and *A. coluzzii*; Burkina Faso, *A. gambiae*, *A. coluzzii* and the *A. gambiae* Goundry form (*Crawford et al., 2016*; *Riehle et al., 2011*), of unknown taxonomic status.

In Kenya, mosquitoes were collected from 10 villages in Nyanza Province near Lake Victoria as described (*Prugnolle et al., 2008*; *Razakandrainibe et al., 2005*). Briefly, indoor-resting, bloodfed females were collected with aspirators from dwellings in the morning during the main rainy seasons of 2002 and 2003 (mid-March to June). Mosquitoes were collected each year for 10 weeks in each of the villages. Following collection, mosquitoes were brought to an insectary, maintained at ambient temperature, humidity, and light with access to sugar solution for 7 days. Mosquito midguts were then dissected to detect and count oocysts. Because mosquitoes were held under controlled conditions for 7-day post-bloodmeal, the design controls for differential mortality of either mosquitoes or parasites based on resting behavior, climatic and temperature variables. This design had the advantage, as compared to the previous study in the same region (*Petrarca and Beier, 1992*) that oocysts were counted directly instead of using CS-ELISA as a measure of infection. Mosquitoes were stored in absolute ethanol.

In the Republic of Guinea (Guinea-Conakry), indoor-resting mosquitoes were collected with aspirators between 06:00 and 12:00 hr in semi-forest (Kenemabomba and Koraboh) and deep forest (Koundara) sites during May 2008. Sites were previously described (*Coulibaly et al., 2016*). Collected mosquitoes were maintained in a field insectary for 48 hr, midguts were dissected, and oocysts were counted microscopically. Carcasses were stored in 80% ethanol at −20 C until DNA extraction. Distinct from the collections in Kenya, these samples were not maintained in the insectary for 7 days.

In Burkina Faso, wild larvae were collected by dipping from breeding sites in the region of Goundry, raised to adults in the insectary, and were fed on gametocytemic blood from naturally infected volunteers from the Goundry vicinity, as previously described (*Riehle et al., 2011*). Unfed mosquitoes were removed, and mosquitoes were dissected 7–8 days post-infection to determine midgut oocyst counts. Membrane feeding sessions resulting from non-infectious or poorly infectious blood were removed using quality filters as defined previously (*Riehle et al., 2011*). In addition, only membrane-feeding experiments that included all three genotypes of the 2La inversion are analyzed. Filtering for infection quality and 2La genotypes yielded six distinct membrane-feeding experiments from the 2008 rainy season, which included the *A. gambiae* complex members, *A. coluzzii*, *A. gambiae* and the Goundry form that types as *A. coluzzii* or *A. gambiae* by standard methods (*Fanello et al., 2002*).

The infection samples were generated by three distinct infection regimes, in order to control for potential biases of any individual method: (i) wild-caught mosquitoes that took a bloodmeal, if at all, under entirely natural conditions and were dissected after holding for 1 week in the insectary (Kenya) to measure infections acquired in nature and developed under controlled conditions, or (ii) wild-caught and dissected within two days of capture (Guinea-Conakry) to measure infections acquired and developed entirely in nature; or (iii) mosquitoes challenged with wild gametocytes by membrane feeding under controlled conditions (Burkina Faso). Exposure of the wild-caught mosquitoes (Kenya and Guinea-Conakry) to parasites occurred, if at all, freely in nature, and thus infection outcome is the combined effect of intrinsic genetic susceptibility plus the influence of factors such as human or animal blood-feeding preference, mosquito age, previous infection history, influence of resting-site preference, and other uncontrolled environmental variables. In contrast, the experimentally infected

samples control for these behavioral and environmental factors that could influence mosquito infection in nature.

## Genotyping for 2La inversion and species

Genomic DNA was extracted from individual female mosquitoes using DNeasy Tissue (Qiagen, CA) or DNAzol (Invitrogen, CA, USA) reagents, and molecular diagnostic tests were used to determine species for all reported samples (*Fanello et al., 2002*). Samples were typed for the 2La inversion using a molecular diagnostic assay (*White et al., 2007*), adapted by using fluorophore-labeled primers and separation by capillary electrophoresis on an ABI 3100 Genetic Analyzer with fragment size calling by Genemapper 4.5 (Applied Biosystems, CA), as described (*Coulibaly et al., 2016*). Primer 23A, spanning the inversion breakpoint, was fluorophore-labeled while the other two primers, DpCross5 and 27A, were unlabeled. The 2La inversion assay yields standard amplicons of 207 bp (2L +$^a$ allele) and 492 bp (2La allele). Non-standard amplicons of 672, 687, 760 and 1020 bp were described, which result from insertion of mobile elements into the breakpoint regions (*Ng'habi et al., 2008*; *Obbard et al., 2007*, *2009*). These larger bands are interpretable derivatives of the standard amplicon sizes and therefore can be assigned to one of the two inversion alleles. All 2La inversion genotyping calls were additionally confirmed by sizing the PCR products by agarose gel electrophoresis. Eight of the 273 Kenyan samples yielded a larger band size (n = 7 for 687 bp and n = 1 for 1020 bp). Of these eight individuals, four were infected and four were non-infected with *P. falciparum*. No samples from Guinea-Conakry or Burkina Faso yielded non-standard band sizes. To be thorough, we carried out an additional infection association analysis after removing these eight individuals, and the results were unchanged. Therefore, the analyses presented for the Kenya site include all 273 samples.

## Tests of 2La inversion genotype association with *Plasmodium* infection levels

For association of 2La inversion genotype with oocyst infection prevalence (as shown in *Figure 2A*), test sets were assembled that generated statistical power to test for association in both phenotypic directions (increased and decreased infection prevalence), and with statistical power for detection of association normalized across geographic study sites and infection methodologies. For the wild-collected samples from Guinea-Conakry and Kenya, test sets were comprised of equal numbers of naturally infected and uninfected mosquitoes. Otherwise, natural infection levels in wild mosquitoes would be too low to robustly test association in a tractable sample size. In the Guinea-Conakry collection, raw *P. falciparum* infection prevalence was 3.3% (for the presence of any parasite stage as detected by molecular assay), and represented completely unfiltered wild-collected samples, in which some mosquitoes did not take any bloodmeal, or did not feed on a gametocyte carrier, or fed too recently to develop oocysts. In the Kenya collection, raw oocyst infection prevalence was 25.5%, and represented wild mosquitoes filtered for the presence of bloodmeal and then incubated in the insectary for 7-day post-capture to allow oocysts, if present, to develop.

For both Guinea-Conakry and Kenya samples, the association test sets include all infected mosquitoes from the collection (i.e., individuals with ≥1 midgut oocysts), and an equal number of randomly chosen uninfected mosquitoes from the same collection (i.e., individuals with zero midgut oocysts). The test sets thus display an average infection prevalence of 50%. This analytical approach normalizes statistical power to query genetic effects in both phenotypic directions across heterogeneous sample sets, and normalizes statistical power to detect an effect across sample sets, which would otherwise display different levels of statistical power if not normalized. In contrast, in the Burkina Faso samples, which were membrane-fed on infectious wild gametocytes and unfed mosquitoes removed, the raw oocyst infection prevalence was 54%, and thus all mosquitoes were directly used as the association test set. All statistical analyses compare samples within a given geographic site, and no comparisons are done across sites.

For association of 2La inversion genotype with oocyst infection intensity (as shown in *Figure 2B*), non-parametric Wilcoxon-Mann-Whitney tests were done in R (*R Core Team, 2012*) comparing oocyst intensity by 2La inversion genotype for all individuals that carried ≥1 midgut oocyst. All analyses were done within a geographic site and no comparisons were carried out across sites.

For analysis of the proportion of each 2La inversion genotype present in the infected and uninfected phenotypic groups (as shown in *Figure 2C*), the same samples as in *Figure 2A* were analyzed except instead of comparing prevalence rates across inversion types, the relative frequencies of 2La inversion genotypes were analyzed in uninfected and infected mosquitoes. Statistical comparisons were done using Chi-Square. As in *Figure 2A and B*, comparisons were done within a geographic site, not across sites.

## Tests of mosquito longevity

Mosquito larvae of *A. gambiae* were collected from natural breeding sites in Luanda Maseno, Kenya, and raised to adults in open semi-field insectaries, where they were exposed to natural temperature and humidity fluctuations. Larvae were provided with food and adults with sugar solution. Adults were maintained in semi-field cages and mortality was monitored twice daily until all individuals were dead. Dead mosquitoes were removed and stored at −20 C until DNA was extracted using DNAzol (Invitrogen, USA) and used for 2La inversion genotyping. Three replicate experiments were analyzed by Kaplan Meier regression (*Kaplan and Meier, 1958*) with log-rank test for difference in longevity between 2La/2La and 2L+$^a$/2L+$^a$ homozygotes. p-Values from individual replicates were combined using the method of Fisher (*Fisher, 1925*).

## Tests of mosquito response to fungal infection

Mosquito larvae of *A. gambiae* were collected and raised as described for the tests of mosquito longevity. For each replicate experiment, fifty 2- to 4-day-old females were exposed to the fungus, *Metarhizium anisopliae* strain ICIPE30 using described methods (*Farenhorst and Knols, 2010*). Briefly, mosquitoes were exposed for 1 hr to filter paper treated with spores formulated in ShellSol T at a surface coverage of $10^{10}$ spores/m$^2$. Control mosquitoes were identically exposed to papers treated with ShellSol T alone. After fungal exposure, the control and infected mosquitoes were maintained in semi-field cages and mortality was monitored twice daily. Dead mosquitoes were removed and stored at −20 C before DNA extraction and genotyping. Longevity was compared between 2La/2La and 2L+$^a$/2L+$^a$ homozygotes as described above.

## Tests for bloodmeal source and adult mosquito spatial behavior

In Guinea-Conakry, collections were made in semi-forest (Kenemabomba and Koraboh) and deep forest (Koundara) sites during May 2008, and in Koraboh and Koundara during October and November 2012. In Koraboh and Koundara, mosquitoes attracted to human hosts were captured by aspirator (human-landing capture, HLC) at night between 18:00 and 06:30 hr. Mosquitoes were sampled by indoor HLC (within dwellings) and outdoor HLC (≥10 m from a dwelling). Indoor-resting mosquitoes were also collected by aspirator between 06:00 and 12:00 hr. All mosquitoes were *A. gambiae*. The bloodmeal source was determined by PCR as previously reported (*Coulibaly et al., 2016*) to compare animal and human bloodmeal rates across 2La inversion genotypes. Bloodmeal sources were tested in wild-caught *A. gambiae* adult mosquitoes carrying visibly detectable blood in Guinea-Conakry (indoor-resting aspirator capture described above n = 126, indoor HLC n = 72, and outdoor HLC n = 12). To assess partitioning of adult mosquito spatial behavior in wild *A. gambiae*, we compared the 2La inversion genotype frequencies of individuals captured by indoor and outdoor HLC. Larvae were sampled from breeding sites in the same villages by dipping with a cup. Larval pools of all known ecological types were sampled within the villages, collecting small numbers of larvae per pool from large numbers of pools, to avoid oversampling siblings and to generate ecologically representative larval samples of the peridomestic village habitat, as previously described (*Gnémé et al., 2013*, *Riehle et al., 2011*). Larvae were immediately stored in 80% ethanol and processed as for adults. Where indicated, p-values from individual sites were combined using the method of Fisher (*Fisher, 1925*).

## Analysis of 2La inversion phylogeny

To address the evolutionary origin of the 2La inversion, we analyzed a subset of the whole genome sequences of *A. gambiae* generated by the *Anopheles gambiae* 1000 (Ag1000) Genomes Consortium (*Miles et al., 2016*), provided by permission, and whole genome sequences of the *A. gambiae* Goundry form from Burkina Faso (*Crawford et al., 2016*). 2La inversion genotypes were assigned

for all Ag1000 Phase I individuals by in silico karyotyping. Chromosome 2L genome sequences were clustered by Principal Components Analysis (PCA). Included in the cluster analysis were internal genotyping standards from previously described wild samples from Burkina Faso (*Markianos et al., 2016*) that had been manually genotyped for the 2La inversion using the standard PCR-based diagnostic test (*White et al., 2007*). These internal genotyping standards included all three genotypes, and their distribution among the three PCA clusters of Ag1000 samples was used to assign 2La inversion genotype calls to the PCA clusters (*Figure 6—figure supplement 1*). Independently, analysis by the Ag1000 Consortium clustered 2L chromosome sequences using the ADMIXTURE program, and assigned genotype calls to the resulting three clusters based on sequence divergence from the *A. gambiae* PEST strain reference genome, which is fixed for 2L+$^a$/2L+$^a$ (*Miles et al., 2016*). Our 2La inversion calls and the Ag1000 calls for the same individuals by independent methods were completely concordant (765/765 matching calls for all three genotypes). The *A. gambiae* Goundry form samples were genotyped for the 2La inversion using the PCR diagnostic assay as described above.

For analysis of evolutionary history of the inversion, only inversion homozygotes were analyzed, to avoid any potential biases that could be introduced by computational phasing methods. A total of 222 whole genome sequences were used from individuals sampled in five countries across Africa in which homozygotes were sympatric within the country sample set (in total, 103 2La/2La and 107 2L+$^a$/2L+$^a$ individuals; Guinea-Conakry n = 18, Guinea-Bissau n = 17, Cameroun n = 98, Kenya n = 22, Uganda n = 55, Burkina Faso Goundry form n = 12). For analysis, 5 Mb windows were identified that (i) were outside the inversion and on another chromosome 3L.15–20 Mb, 3R.15–20 Mb, or (ii) were outside the inversion but on the same chromosome 2L.8–13 Mb, or (iii) spanned the proximal and distal breakpoints of the inversion, 2L.18–23 Mb, and 2L.39–44 Mb, or (iv) were entirely within the inversion 2L.29–34 Mb. Using VCF files for these defined genomic regions, maximum likelihood trees were constructed to examine the evolutionary relationship of windows across all 222 individuals. The sequences with degenerate UIPAC code were derived from the VCF files for these defined genomic regions, and phylogenetic trees were inferred by the maximum likelihood approach using the GTR+-Gamma substitution model (*Yang, 1994*) implemented in IQ-TREE version 1.5.4 (*Nguyen et al., 2015*). The figures were made with the R package ape (*Paradis et al., 2004*). Phylogenetic tree branches are colored to reflect geographic origin and labeled with 2La inversion karyotype (label at branch tip, black a, 2La/2La and red +, 2L+$^a$/2L+$^a$).

## Correlation between 2La inversion allele frequency and annual precipitation

Field collection of 2La allele frequency data in *A. gambiae* and *A. coluzzii* in Guinea-Conakry and southern Mali was previously described (*Coulibaly et al., 2016*). Briefly, mosquitoes were sampled at sites over a 350 km transect representing ecological zones of arid savanna (Takan, southern Mali, n = 190, overall 2L+$^a$ frequency of 3.7%; n = 165 *A. coluzzii* (95% 2La/2La, 5% 2La/2L+$^a$) and n = 25 *A. gambiae* (84% 2La/2La, 12% 2La/2L+$^a$ and 4% 2L+$^a$/2L+$^a$), humid savanna (Toumani Oulena, southern Mali, n = 191, overall 2L+$^a$ frequency of 13.9%, n = 23 *A. coluzzii* (91% 2La/2La and 9% 2La/2L+$^a$) and n = 168 *A. gambiae* (73% 2La/2La, 23% 2La/2L+$^a$, and 4% 2L+$^a$/2L+$^a$), semi-forest (Koraboh, Guinea-Conakry, n = 366, overall 2L+$^a$ frequency of 53.8%, all *A. gambiae* (29% 2La/2La, 36% 2La/2L+$^a$ and 35% 2L+$^a$/L+$^a$) and deep forest (Koundara, Guinea, n = 259, overall 2L+$^a$ frequency of 59%, n = 16 *A. coluzzii* (19% 2La/2L+$^a$ and 81% 2L+$^a$/2L+$^a$) and n = 243 *A. gambiae* (34% 2La/2La, 25% 2La/2L+$^a$, 41% 2L+$^a$/2L+$^a$). Annual precipitation data for collection locations with a resolution of ~1 km$^2$ were extracted from the WorldClim (version 1.4) current global bioclimatic spatial database (*WorldClim, 2005*; *Hijmans et al., 2005*). All computations were done using R (*R Core Team, 2012*).

## Ethical considerations

For collection of blood from *P. falciparum* gametocyte carriers for experimental membrane feeder infection of mosquitoes, the study protocol was reviewed and approved by the institutional and national health ethical review board (Commission Nationale d'Ethique en Santé) of Burkina Faso (code N° 2006–032). The study procedures, benefits and risks were explained to parents or legal guardians of children and their informed consent was obtained. Children of parents or guardians

who had given consent were brought to CNRFP the day of the experiment for gametocyte carrier screening. All children were followed and symptomatic subjects were treated with the combination of artemether-lumefantrine (Coartem) according to relevant regulations of the Burkina Faso Ministry of Health.

For human-landing capture of mosquitoes, the study protocol was reviewed and approved by the institutional and national health ethical review boards (Comité National d'Ethique pour la Recherche en Santé) of The Republic of Guinea (reference number N'003/CNFRSR/MAF/13) and (Comité d'Ethique Institutionelle/FMPOS) of the Republic of Mali (reference number Traore-37). The study procedures, benefits and risks were explained to collectors and their informed consent was obtained. All collectors were followed and symptomatic subjects were treated with the combination of artemether-lumefantrine (Coartem) according to relevant regulations of the Republic of Guinea and Republic of Mali Ministries of Health.

## Acknowledgements

We thank Dr Jean N K Maniania (Arthropod Pathology Unit, ICIPE, Nairobi Kenya) for the supply of fungal spores, and Julien Guglielmini (C3BI, Institut Pasteur, Paris) for advice on phylogenetic analysis. This work received financial support to KDV from the European Commission, FP7 Infrastructures #228421 Infravec; European Commission, Horizon 2020 Infrastructures 731060 Infravec2; European Research Council, Support for frontier research, Advanced Grant #323173; National Institutes of Health, NIAID R01 #AI073685; and Agence Nationale de la Recherche Laboratoire d'Excellence 'Integrative Biology of Emerging Infectious Diseases' #ANR-10-LABX-62-IBEID, to MMR from National Institutes of Health, NIAID R21 #AI121587. The funders had no role in study design, data collection and analysis, decision to publish, or preparation of the manuscript.

## Additional information

### Funding

| Funder | Grant reference number | Author |
| --- | --- | --- |
| European Commission | 323173 | Kenneth D Vernick |
| National Institutes of Health | AI121587 | Michelle M Riehle |
| European Commission | 228421 | Kenneth D Vernick |
| European Commission | 731060 | Kenneth D Vernick |
| Agence Nationale de la Recherche | ANR-10-LABX-62-IBEID | Kenneth D Vernick |

The funders had no role in study design, data collection and interpretation, or the decision to submit the work for publication.

### Author contributions

MMR, KDV, Conceptualization, Data curation, Formal analysis, Supervision, Funding acquisition, Investigation, Methodology, Writing—original draft, Project administration, Writing—review and editing; TB, WMG, Conceptualization, Data curation, Supervision, Investigation, Methodology, Writing—original draft, Project administration, Writing—review and editing; AG, AP, EB, FR, Investigation, Methodology, Formal Analysis, Writing—review and editing; BC, AF, Conceptualization, Investigation, Methodology, Writing—review and editing; AHB, N'FS, Supervision, Methodology, Writing—review and editing; SFT, Supervision, Investigation, Methodology, Writing—review and editing

### Author ORCIDs

Michelle M Riehle, http://orcid.org/0000-0003-1042-1653
Kenneth D Vernick, http://orcid.org/0000-0003-4336-312X

## Ethics

Human subjects: For collection of blood from P. falciparum gametocyte carriers for experimental membrane feeder infection of mosquitoes, the study protocol was reviewed and approved by the institutional and national health ethical review board (Commission Nationale d'Ethique en Santé) of Burkina Faso (code N° 2006-032). The study procedures, benefits and risks were explained to parents or legal guardians of children and their informed consent was obtained. Children of parents or guardians who had given consent were brought to CNRFP the day of the experiment for gametocyte carrier screening. All children were followed and symptomatic subjects were treated with the combination of artemether-lumefantrine (Coartem) according to relevant regulations of the Burkina Faso Ministry of Health. For human-landing capture of mosquitoes, the study protocol was reviewed and approved by the institutional and national health ethical review boards (Comité National d'Ethique pour la Recherche en Santé) of The Republic of Guinea (reference number N'003/CNFRSR/MAF/13) and (Comité d'Ethique Institutionelle/FMPOS) of the Republic of Mali (reference number Traore-37). The study procedures, benefits and risks were explained to collectors and their informed consent was obtained. All collectors were followed and symptomatic subjects were treated with the combination of artemether-lumefantrine (Coartem) according to relevant regulations of the Republic of Guinea and Republic of Mali Ministries of Health.

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
