## [Decision Letter]

Thank you for submitting your article "The *Anopheles gambiae* 2La chromosome inversion is associated with susceptibility to *Plasmodium falciparum* in Africa" for consideration by *eLife*. Your article has been favorably evaluated by Prabhat Jha (Senior Editor) and three reviewers, one of whom served as Guest Reviewing Editor. The reviewers have opted to remain anonymous.

The reviewers have discussed the reviews with one another and the Reviewing Editor has drafted this decision to help you prepare a revised submission.

Summary:

The study presented by Riehle, et al., focuses on a known chromosomal inversion polymorphism present in some species in the *Anopheles gambiae* species complex, a major contributor of malaria transmission in Africa. Using collections made across the African continent comprised of 2La/2La, 2L+^a^/2L+^a^, and heterozygous individuals, they explore the association of the 2La genotype with several components of vectorial capacity that would lead to asymmetric malaria transmission among individuals with different 2La genotypes. Specifically, the authors find that the 2La inversion is associated with intrinsic Plasmodium susceptibility and exophilic feeding behavior, but not longevity, human feeding rate, or immune function. The reviewers agree this report sheds light on how polymorphisms within the species complex affect transmission dynamics and, by extension, malaria epidemiology.

Essential revisions:

The reviewers raise several issues that must be adequately addressed before they can accept the paper for publication. Few, if any of the requested revisions require additional data collections but rather different analysis and additional justification/discussion.

1) The paper currently reads as several component parts rather than a unified story. In particular, the initial experiment (Figure 2) identifies an intrinsic difference in Pf susceptibility but, rather than explore the mechanism for that, subsequent experiments explore additional behavioral differences. Please revise the paper structure to present a unified study.

2) The samples from west Africa appear to be pooled from all taxa, potentially including *A. gambiae, A. coluzzii*, the Goundry form and/or *A. gambiae/A. coluzzii* hybrids. Please provide data on the relative abundance of each of these taxa in the west African samples and the frequencies of the 2La genotype in each.

3) While infection prevalence may be the most important metric to determine infectiousness, the reviewers agree oocyst intensity data provides necessary insight into pathogen susceptibility. Please include it.

4) Data collection for Figure 2 is very convoluted as each collection site had a different way of assessing infection and prevalence, yet the figure presents as if they were treated identically. For example, the Kenyan mosquitoes have a much greater chance of being considered infected via oocyst counting than the Guinea-Conakry due to the longer incubation time. The artificial 50% prevalence set for both of these groups is also deceiving. The data could be better described as proportions of each haplotype in infected and uninfected groups rather than true prevalence. Either way, the inconsistencies should be up front in the text and not just the Methods.

5) Justify why exposure to a single fungal species is representative of "generalized pathogen susceptibility" and immunity to *P. falciparum*.

6) For Figure 5, discuss the possibility that the 2L+^a^/2L+^a^ individuals may be feeding on animals and resting outdoors, so not represented in your collections. This outcome would alter the conclusions.

7) Following Comment 1, integrate results from Figure 7 and 8 into the Results section and discuss in the Discussion section.

8)) Some limitations to the molecular karyotyping of the 2La inversion have been published (Ng'habi KR, et al. (2008) Parasites & Vectors 1:45), This should be mentioned and a discussion of why the authors have confidence in this method provided.

9) The references section needs to be restructured. While referred to in the article by author name, the order is shown by numbers in order of appearance.

---

## [Author Response]

*Essential revisions:*

*The reviewers raise several issues that must be adequately addressed before they can accept the paper for publication. Few, if any of the requested revisions require additional data collections but rather different analysis and additional justification/discussion.*

*1) The paper currently reads as several component parts rather than a unified story. In particular, the initial experiment (Figure 2) identifies an intrinsic difference in Pf susceptibility but, rather than explore the mechanism for that, subsequent experiments explore additional behavioral differences. Please revise the paper structure to present a unified study.*

We modified the text throughout to unify the scientific narrative.

We also modify the text to clarify insight derived about the potential mechanisms underlying the inversion 2La-associated infection phenotype. Briefly, we find that the infection association is not a local phenomenon, but is Africa-wide, consistent with the single monophyletic origin of the 2La inversion across geography and *A. gambiae* taxa. Consequently, the differential infection phenotype is probably a shared, common mechanism. The mechanism is not due to differential human blood feeding or survival, and is not a consequence of overall immune competence. Thus, the differential infection is relatively specifically addressed to *P. falciparum*. These mechanistic observations are confirmed by the detection of the association in wild-caught adult mosquitoes, which combine together the effects of survival, immunity, and bloodfeeding behavior, which we also dissect as separate tests of mechanism. Moreover, these observations are further confirmed and extended by detection of the association in controlled experimental infections. Taken together, these results provide considerable information about the mechanism of the trait, and they all point in the same direction: towards a genetically-controlled physiological difference in 2La inversion genotypes that underlie differences in parasite development efficiency.

The genetically-controlled difference must have a cellular and molecular basis. Functional dissection of a molecular mechanism is beyond the scope of a field-based study like the current one. However, some mosquito genetic loci and candidate genes involved in malaria susceptibility are located in the 2La inversion, and we discuss their potential to be involved in the 2La inversion association. We also discuss the main limitation involved in genetically resolving the trait, which is the extent of linkage across the 22 Mb 2La inversion. Because there is almost no recombination between alternate forms, the haplotypes probably comprise supergenes, and it is likely that multiple synergistic gene products together underlie the phenotype. Thus, as we now clarify, a single “locus” or gene explaining the mechanism is not likely to exist, because the phenotype probably results from the synergistic effect of multiple/many genes dispersed throughout the 22 Mb haplotype.

*2) The samples from west Africa appear to be pooled from all taxa, potentially including A. gambiae, A. coluzzii, the Goundry form and/or A. gambiae/A. coluzzii hybrids. Please provide data on the relative abundance of each of these taxa in the west African samples and the frequencies of the 2La genotype in each.*

Multiple taxa were present in collections used for two analyses: infection association in Burkina Faso (Figure 2), and ecological correlation in Guinea and Mali (Figure 7). The other collections used, Guinea infection samples (Figure 2) and behavior samples (Figure 5) and infection samples from Kenya (Figure 2) were all *A. gambiae*.

In all cases, the collections used for the different analyses are representative of the taxa that were present at the study sites, and no sample sets were filtered for taxa. The breakdown of taxa composition and allele frequencies for the above two analyses with multiple taxa are now added to the manuscript as follows.

Figure 2 – Burkina Faso, infection association. Samples were comprised of *A. gambiae, A. coluzzii*, and the Goundry form. We generated a new figure supplement to Figure 2 (Figure 2—figure supplement 2) that indicates the numbers of each taxon and frequencies of 2La genotypes. The same association of 2L+^a^ and infection levels as seen in the Burkina Faso pooled samples (Figure 2) is also consistent on a fine scale in the breakdown of the figure supplement. Specifically, in all pairwise comparisons, 2L+^a^ is significantly more susceptible than 2La from any taxon, either within Goundry, or as compared to 2La of *A. gambiae* or *A. coluzzii*.

Figure 7 – Guinea and Mali, ecological correlation. Samples were comprised of *A. gambiae* and *A. coluzzii*. Numbers of each taxon and 2La frequencies per study site are now listed in the Methods section, “Correlation between 2La inversion allele frequency and annual precipitation”.

In addition, we used new whole-genome sequence data from Goundry mosquitoes to generate a new phylogenetic tree of the 2La inversion including Goundry. Figure 6 is now revised to include Goundry mosquitoes. This analysis now extends the observation of monophyletic ancestry of the 2La inversion to also include the 2La alleles segregating within the Goundry form. The new tree demonstrates that 2La inversion alleles share a common evolutionary history that is deeper than taxonomic group or geography, even to the Goundry form.

*3) While infection prevalence may be the most important metric to determine infectiousness, the reviewers agree oocyst intensity data provides necessary insight into pathogen susceptibility. Please include it.*

We now include data on oocyst infection intensity (new Figure 2). This analysis shows that the 2L+^a^ inversion allele is significantly associated with greater infection intensity in the experimentally infected Burkina Faso samples (new Figure 2), in addition to with higher infection prevalence as shown in the manuscript (Figure 2). The same trend for infection intensity, although non-significant, possibly due to smaller sample size, is also seen in the Kenyan population. The Guinea samples did not have statistical power for analysis of infection intensity because of small sample size, and because all 7 2La/2La homozygotes were uninfected. Infection intensity is analyzed using only individuals carrying ≥1 oocyst, to avoid confounding effects of intensity and prevalence.

*4) Data collection for Figure 2 is very convoluted as each collection site had a different way of assessing infection and prevalence, yet the figure presents as if they were treated identically. For example, the Kenyan mosquitoes have a much greater chance of being considered infected via oocyst counting than the Guinea-Conakry due to the longer incubation time. The artificial 50% prevalence set for both of these groups is also confusing. The data could be better described as proportions of each haplotype in infected and uninfected groups rather than true prevalence. Either way, the inconsistencies should be up front in the text and not just the Methods.*

As suggested by the reviewers, we now also present the proportions of each 2La inversion haplotype in the infected and uninfected groups (new Figure 2). As suggested by the reviewers, this alternative analysis does not rely on infection prevalence. All of these results are significant, in the same direction, and with similar p-values, as the original figure (the prevalence analysis, now renamed Figure 2). This alternative analysis using haplotype and infection strengthens the association result, and we thank the reviewers for the suggestion.

Regarding the association analysis by infection prevalence in current Figure 2 (old Figure 2), we previously described the analytical method with extensive detail in Methods. However, we now add text also in Results to clarify the analytical approaches used. By constituting association test sets with all infected mosquitoes from each collection, and an equal number of randomly chosen uninfected mosquitoes from the same collection, “the test sets display an average infection prevalence of 50%, and therefore have the power to detect genetic association for either increased or decreased infection prevalence.” This analytical approach normalizes statistical power to query genetic effects in both phenotypic directions across heterogeneous sample sets, which would otherwise display different levels of statistical power if not normalized.

Regarding the different infection methods used at the different study sites, they are not “inconsistencies” but were purposely included to augment the power of the study to query different physiological aspects of infection. However, the design is complex and we agree that it could be confusing for the reader. We now add additional explanatory text in Results to clarify the infection methodologies, and to clarify that observing the same empirical results under three distinct experimental regimes reinforces the robust nature of the findings. Conversely, applying a single homogeneous infection design would have been a less stringent test, and would have generated less information about the phenotype and its potential mechanisms (for example, behavioral versus physiological).

Finally, we clarify with additional text in Methods and Results that all comparisons are made between 2La genotypes *within* a geographic location, and no comparisons are made *across* sites. Put differently, analysis of 2La association with infection were made among sympatric samples using the same infection methodology, that is, within Kenya samples, or within Guinea samples, but not between Kenya and Guinea.

*5) Justify why exposure to a single fungal species is representative of "generalized pathogen susceptibility" and immunity to P. falciparum.*

One possible explanation for the greater *P. falciparum* infection levels of 2L+^a^ carriers could be that these individuals display a low overall level of immune competence, generalized across all of their antimicrobial defenses. In other words, perhaps they are highly infected with malaria not as a specific effect, but because they have poor defenses against all infections. We now clarify in the text that the longevity test (Figure 3) comprised a comprehensive natural test for 2La genotype differences in immune competence to natural microbes and pathogens. In this test, mosquitoes grown from wild-caught larvae in Kenya were kept in outdoor open cages, fully exposed to all environmental microbes and pathogens. There was no difference in survival among 2La inversion genotypes, suggesting that the greater *P. falciparum* infection levels of 2L+^a^ carriers is not due to a general low immune function or competence.

However, because we could not be certain that the ambient natural microbes in the Figure 3 test included any highly virulent pathogens, we decided to employ a known virulent Anopheles pathogen, the entomopathogenic fungus, Metarhizium, in a more stringent assay for survival difference between 2La inversion genotypes. Entomopathogenic fungi are model eukaryotic pathogens commonly used in studies of mosquito immunity, as well as a proposed biopesticide for vector control. The virulence of Metarhizium as a pathogen was confirmed by the elevated mortality effect, but the mortality curves were not different between 2La genotypes (Figure 4). Of course, Metarhizium is just one eukaryotic pathogen, as pointed out by the reviewers. However, the two results taken together indicate that the *P. falciparum* susceptibility of 2L+^a^ carriers is more specific than, and cannot be explained by, generalized immune incompetence.

*6) For Figure 5, discuss the possibility that the 2L+a/2L+^a^ individuals may be feeding on animals and resting outdoors, so not represented in your collections. This outcome would alter the conclusions.*

We added new text in the Discussion on this point. In Results, we reported a deficit of 2L+^a^/2L+^a^ captured as HLC adults as compared to 2L+^a^/2L+^a^ among larvae (2L+^a^/2L+^a^ were 27% of total HLC versus 42% of total larvae). Nevertheless, a large fraction of 2L+^a^/2L+^a^ were capturable. Among these capturable 2L+^a^/2L+^a^, the rate of human bloodmeal carriage was 100% (Results), and the level of *P. falciparum* infection was significantly elevated as compared to 2La/2La (Figure 2). The feeding behavior of the noncaptured or “missing” fraction of 2L+^a^/2L+^a^ cannot be determined directly, and it cannot be ruled out that they could be less anthropophilic. However, it is not likely that the missing 2L+^a^/2L+^a^ display a radically different (i.e., zoophilic) behavior than the capturable ones, for several reasons. First, most knowledge of *A. gambiae/coluzzii* behavior to date indicates strong human-feeding propensity, with very few accounts of zoophily, so an exclusively zoophilic group within *A. gambiae* would be unexpected. Second, 2L+^a^ carriers were previously described as more exophilic (Coluzzi et al. 1979). Third, mosquitoes of the Forest Form of *A. gambiae*, which are 2L+^a^/2L+^a^, are both exophilic and anthropophilic (Bockarie et al. 1993, Bockarie et al. 1994).

Thus, the missing 2L+^a^/2L+^a^ in our study most likely indicate a behavioral *tendency* towards exophily of the anthropophilic 2L+^a^/2L+^a^. In other words, the missing fraction represent one tail of an overall 2L+^a^/2L+^a^ distribution that is skewed towards exophily, while 2La/2La behavior is skewed towards endophily, and both are likely equally anthropophilic. It is this behavioral tendency of 2L+^a^/2L+^a^ that makes them a potential threat, because they could respond in an exophilic direction when placed under selective pressure by ITNs and IRS, consistent with empirical data from ITN distribution in Kenya (Matoke-Muhia et al., 2016).

*7) Following Comment 1, integrate results from Figure 7 and 8 into the Results section and discuss in the Discussion section.*

The old Figure 7 has been integrated into the Results (new Figure 7), and is discussed in the Discussion section. The old Figure 8 is now a figure supplement to Figure 6 (Figure 6—figure supplement 1).

*8) Some limitations to the molecular karyotyping of the 2La inversion have been published (Ng'habi KR, et al. (2008) Parasites & Vectors 1:45), This should be mentioned and a discussion of why the authors have confidence in this method provided.*

We added new text in Methods on this point. Ng’habi et al. (2008), Obbard et al., (2007) and Obbard et al. (2008) point out larger band sizes for the standard PCR based genotyping assay for the 2La inversion. These larger bands are a result of insertion/deletion derivatives of the standard PCR amplicon sizes 207 bp (expected size for 2L+^a^) and 492 bp (expected sized for 2La), and therefore can be assigned to an inversion form (2La or 2L+^a^). To survey for any amplicon sizes outside of the expected 207 bp and 492 bp, the 2La genotyping PCR reactions of all samples reported were amplified using a fluorescent primer and size-separated by capillary electrophoresis on an ABI sequencing machine, as well as verified on agarose gels. For the Kenyan samples, 8 of 273 samples generated non-standard band sizes (n=7 for 687 bp and n=1 for 1020 bp). Of these 8 samples, 4 were infected and 4 were non-infected. To be thorough, we ran an additional infection association analysis after removing these 8 individuals, and the results were unchanged. Therefore, the analyses presented include all samples.

*9) The references section needs to be restructured. While referred to in the article by author name, the order is shown by numbers in order of appearance.*

The references have been reformatted using the Chicago style, now appearing by author and date in the text, and by alphabetical order in the bibliography.